# Impact of Thermal Manipulation of Broiler Eggs on Growth Performance, Splenic Inflammatory Cytokine Levels, and Heat Shock Protein Responses to Post-Hatch Lipopolysaccharide (LPS) Challenge

**DOI:** 10.3390/ani15121736

**Published:** 2025-06-12

**Authors:** Mohammad Borhan Al-Zghoul, Seif Hundam, Mohammad Mayyas, David E. Gerrard, Rami A. Dalloul

**Affiliations:** 1Department of Basic Medical Veterinary Sciences, Faculty of Veterinary Medicine, Jordan University of Science and Technology, Irbid 22110, Jordan; sfhundam23@vet.just.edu.jo; 2Department of Animal Production, Faculty of Agriculture, Jordan University of Science and Technology, Irbid 22110, Jordan; mohammad.mayyas@jsmo.gov.jo; 3School of Animal Sciences, Virginia Polytechnic Institute and State University, Blacksburg, VA 24061, USA; dgerrard@vt.edu; 4Department of Poultry Science, University of Georgia, Athens, GA 30602, USA; rami.dalloul@uga.edu

**Keywords:** broiler, cytokines, immune response, lipopolysaccharide, thermal manipulation

## Abstract

Raising healthy chickens without relying on antibiotics is vital for food safety and animal welfare. One promising approach is to gently heat eggs during a specific stage of their development, a process called thermal manipulation. This study tested whether this method helps chickens respond better to lipopolysaccharide (LPS) challenge after hatching. We exposed eggs to slightly elevated temperatures for several hours each day during a crucial development period. After hatching, some chickens were given LPS to mimic bacterial infection. We then measured their bodies’ reactions, including changes in weight, temperature, and mRNA expression of immune-related genes. We found that while thermal manipulation did not affect hatchability, it shortened the time needed to hatch and improved the efficiency with which chickens used their feed. Importantly, the treated chickens had a more balanced immune response, showing fewer signs of excessive inflammation and better stress control. This suggests that heat conditioning during embryo development can help chickens effectively fight off infections and recover more quickly. These findings offer an antibiotic-free approach to support poultry health, benefiting farmers, consumers, and public health.

## 1. Introduction

Broiler chickens are a globally significant livestock species that play a crucial role in human nutrition [1]. However, bacterial infections cause a serious challenge in the poultry industry and considerably threaten chickens, leading to substantial economic losses [2]. Due to the substantial impact of bacterial infections on poultry production, a comprehensive strategy is essential to enhance broiler chickens’ immune systems and overall health. This includes regularly administering antibiotics, which is a crucial component of this approach [3].

The widespread use of antibiotics for therapeutic and prophylactic purposes in poultry production has led to the emergence of antibiotic resistance in commensal bacteria (e.g., *Escherichia coli*, *enterococci*) and zoonotic enteropathogens (e.g., *Salmonella*, *Campylobacter*) [4]. Beyond fostering resistance, the excessive administration of antibiotics results in the accumulation of antibiotic residues in broiler tissues. These residues persist in broiler meat, posing a potential risk to human consumers through chronic low-dose exposure. Such exposure may further drive the development of antibiotic resistance in human pathogens, exacerbating a critical public health concern [5].

TM, exposure to moderate heat stress during broiler chickens’ key embryonic development phase has enhanced tissue stability, oxidative stress response immune response during heat stress, and bacterial challenge with *E. coli* [6,7]. Therefore, TM is proposed as an alternative method to enhance chickens’ reactions to stressful situations. Indeed, in broiler chickens, many studies have shown TM’s ability to improve thermotolerance acquisition, hatchability, body weight, and immune response to heat stress [7,8,9]. Furthermore, it was reported that the TM of Pekin duckling embryos significantly reduced bursal and splenic HSP70 and IL-6 mRNA levels after the LPS treatment challenge [10].

Inflammation is a critical host response to infection, mediated by immune cells such as monocytes and macrophages, which release proinflammatory cytokines (e.g., tumor necrosis factor α (TNF-α), interleukins (IL-1β, IL-6, and IL-8)) upon pathogen recognition [11]. Similarly, heat shock proteins (HSPs), including HSP40, HSP60, HSP70, and HSP90, play essential roles in cellular stress responses by maintaining protein integrity and preventing misfolded protein aggregation [12]. We previously demonstrated that different TM methods during broiler chicken embryogenesis led to significant alterations in the levels and dynamics of HSP and cytokine mRNA expressions, which correlated with enhanced thermotolerance during subsequent heat stress in broiler chickens [13].

However, limited data are available regarding the impact of TM during the embryonic period on the immune response to LPS challenge. Therefore, this study aimed to evaluate the effects of TM during embryogenesis, alongside LPS challenge, on body temperature (Tb), body weight (BW), and splenic mRNA expressions of proinflammatory cytokines (IL-1β, IL-2, IL-6, IL-8, TNF-α, IL-12, and IFN-γ), anti-inflammatory cytokines (IL-10 and TGF-β), Toll-like receptors (TLR-1-4), transcription factor proteins (nuclear factor kappa B (NF-κB) and NF-κB p50), heat shock proteins (HSP70 and HSP 90), and heat shock factors (HSF-1 and HSF-3).

## 2. Materials and Methods

The Animal Care and Use Committee at Jordan University of Science and Technology (JUST-ACUC; approval #16/3/2/205) authorized all the experiments, management conditions, and techniques used in this study.

### 2.1. Experimental Design and Animal Management

The current study’s experimental design is summed up in Figure 1. The study was conducted in two sequential phases, examining TM during embryogenesis and its subsequent effects on post-hatch immune challenge responses. In the first phase, fertile eggs (*n* = 740) from 35-week-old Indian River breeders were obtained from certified distributors in Ar-Ramtha, Jordan. After excluding damaged or abnormally sized eggs (outside the 55–70 g range), selected eggs averaging 62 ± 3 g (*n* = 720) were randomly divided between the control and TM groups. Control eggs were incubated under standard conditions (37.8 °C, 56% RH throughout embryogenesis), while the TM group eggs received elevated temperatures (38.8 °C with 65% RH for 18 h daily) specifically during embryonic days 10–18. Spot candling was performed on day 7 using a smartphone LED flashlight (Samsung Galaxy A32, Samsung Electronics Co., Ltd., Suwon, The Republic of Korea) to identify and remove infertile eggs and those containing non-viable embryos. RH was maintained using the incubator’s integrated automatic humidity regulation system. The system operated via a water reservoir and internal sensors that continuously monitored and adjusted RH levels to maintain the desired setpoint.

### 2.2. Post-Hatch Rearing and Growth Performance Monitoring

Following hatching, broiler chicks were transferred to the Animal House at JUST for the second experimental phase. Chicks were feather-sexed (47.3% male, 52.7% female) with males distributed into 16 cages per treatment (80 cm × 40 cm × 100 cm, 8–10 birds/cage). All birds received NRC-recommended basal diets in pellet form without antibiotics or vaccinations (Table 1) [14], with ad libitum access to feed and water. Birds were housed in a closed-house system equipped with environmental controls to maintain stable conditions. During the experimental period, a continuous 24 h lighting program was implemented for all broiler chicks to ensure optimal feed intake and uniform growth conditions. This lighting regime was consistently maintained throughout the study for all treatment groups. The room temperature was maintained at 35 ± 1 °C for the first three days and then gradually reduced to 20 °C by the fourth week. Weekly body weights and daily feed intake were recorded to calculate growth performance metrics. No antibiotics or vaccinations were administered during the study. All birds were raised under similar management practices.

Birds were individually weighed (*n* = 10 per cage) weekly from the first to the fifth week of age. The change in body weight was calculated by subtracting the initial average live weight for a specific period from the average final live weight during the same experimental period. Feed intake was recorded daily based on the amount consumed per cage. The average daily gain (ADG), daily feed intake (DFI), and feed conversion ratio (FCR) were determined at the pen level for each feeding phase (days 1–7, 8–14, 15–21, 22–28, and 29–35), as well as for the overall experimental period (days 1–35). The experimental trial lasted for 35 days. The room temperature was maintained at 35 ± 1 °C during the first three days, then gradually reduced to 20 °C by the end of the fourth week. Standard feed rations and water were provided to the chicks ad libitum during the field experiment.

### 2.3. LPS Challenge (Phase 2)

In the second experimental phase, we evaluated the effects of embryonic thermal conditioning on post-hatch immune responses. At 15 days of age, broiler chickens from both the control and TM groups were randomly divided into two subgroups (*n* = 80 per subgroup) for immune challenge. The first subgroup received an intraperitoneal injection of 0.5 mL 0.9% normal saline (vehicle control), while the second subgroup was administered 0.5 mL of *Escherichia coli O127* LPS (0.5 mg/kg). This 2 × 2 factorial design (control vs. TM × saline vs. LPS) allowed for a comprehensive assessment of both the independent and interactive effects of embryonic TM and post-hatch immune challenge.

Immediately following injection, all chickens were transferred to a dedicated experimental room maintained under thermoneutral conditions to eliminate confounding environmental stressors. Throughout the 6 h observation period, mortality was systematically recorded. Physiological responses were quantified by measuring core body temperature and body weight at 1, 3, and 6 h post-injection using a J/K/T thermocouple meter (Kent Scientific Corp., Torrington, CT, USA) with a rat rectal probe (±0.1 °C accuracy).

### 2.4. RNA Isolation and cDNA Synthesis

At 1, 3, and 6 h post-injection, ten broiler chickens per subgroup (40 total per time point) were randomly selected through a blinded manual selection process from their housing cages and humanely euthanized following administration of either LPS or saline. Spleen tissues were aseptically collected and rapidly frozen on-site with liquid nitrogen to avert RNA degradation. The samples were preserved in TRI Reagent^®^ solution tubes (Zymo Research Co., Irvine, CA, USA) at −20 °C. Tissues were homogenized with a Bead Ruptor Elite-Bead Mill Homogenizer (OMNI International, Kennesaw, GA, USA). Total RNA was extracted from splenic samples utilizing a Direct-Zol™ RNA MiniPrep (Zymo Research Co., Irvine, CA, USA) in conjunction with TRI Reagent^®^ (Zymo Research Co., Irvine, CA, USA). The RNA’s amount and quality were assessed utilizing a Qubit 4 Fluorometer (Thermo Fisher Scientific, Waltham, MA, USA), a Biotek PowerWave XS2 Spectrophotometer (BioTek Instruments, Inc., Winooski, VT, USA), and a 1% agarose gel. cDNA was generated for each sample with PrimeScript™ RT Master Mix (Zymo Research Co., Irvine, CA, USA), utilizing 500 ng of RNA per reaction.

### 2.5. Relative-Quantitative Real-Time PCR (RT-qPCR)

The Blastaq™ Green qPCR Master Mix (Applied Biological Materials Inc., Richmond, BC, Canada) was employed using Rotor-Gene Q MDx 5 plex equipment (Qiagen, Hilden, Germany). A 20 µL reaction mixture was produced by amalgamating 10 µL of master mix, 2 µL of forward primer (2 pmol), 2 µL of reverse primer (2 pmol), 2 µL of sample cDNA, and 4 µL of nuclease-free water. The cycling conditions included 50 °C for 2 min, 95 °C for 15 min, followed by 40 cycles of 95 °C for 10 s, 57 °C for 30 s, and 72 °C for 10 s, culminating in a final melting step at 95 °C for 20 s. Replicates of each cDNA sample were examined, and fluorescence emission was observed. Relative quantification was computed automatically based on the fluorescence signals. β-Actin and Glyceraldehyde-3-Phosphate Dehydrogenase (GAPDH) functioned as internal controls for normalizing the fold variations in gene expression. The melting curve confirmed the specificity of single-target amplification.

The cDNA sequence for each gene was retrieved from NCBI’s Nucleotide database (https://www.ncbi.nlm.nih.gov/nucleotide/, accessed on 9 June 2025). All primers were created via IDT Primer Quest software (http://eu.idtdna.com/PrimerQuest/Home/Index, accessed on 9 June 2025). The primer sequences are presented in Table 2.

### 2.6. Organ Weights

On day 35 post-hatch, 10 broilers were randomly selected and individually weighed. The broilers were humanely euthanized via cervical dislocation and promptly necropsied. We recorded absolute and relative weights (expressed as a percentage of live body weight) for all major organs, including liver, spleen, heart, gizzard, proventriculus, gallbladder, and intestinal segments. The lengths of the small and large intestines were measured using a flexible tape and a metal ruler to ensure accuracy.

### 2.7. Statistical Analysis

All statistical analyses were conducted using IBM SPSS Statistics 27.0 (IBM Software, Chicago, IL, USA). The hatchability parameters were assessed with the chi-square test due to the binary outcome nature of the data. Continuous variables—including body temperature (Tb), body weight (BW), and mRNA gene expression levels—were expressed as means ± standard deviation (SD). These variables were analyzed using a two-way factorial ANOVA (2 × 2 design), with thermal manipulation (TM vs. control) and post-hatch injection (LPS vs. saline) as the two fixed factors. Hatch time, recorded as the number of hours from egg setting to external pip formation for each chick, was compared between groups using Student’s *t*-test after verifying normality assumptions.

For gene expression analysis, Ct values were normalized against two reference genes (β-actin and GAPDH), and relative expression was calculated using the 2^−ΔΔCt^ method. Before analysis, data were examined for normal distribution and homogeneity of variance. Post hoc comparisons were performed using Tukey’s HSD test when significant main or interaction effects were observed. Statistical significance was set at *p* < 0.05 for all analyses.

### 2.8. Use and Potential Use of AI

While preparing this work, the authors used QuillBot and Grammarly (version: 1.2.149.1641) to improve readability and check grammar and spelling. After using these tools, the authors reviewed and edited the content as needed.

## 3. Results

### 3.1. The Impact of TM on Hatchability Rate, Body Weight (BW), Body Temperature (Tb), Growth Performance Metrics, and Internal Organ Weight

TM during embryogenesis did not significantly alter the hatchability rate compared to the control group (93.71 ± 1.09% vs. 93.03 ± 0.54%, *p* > 0.05). No differences were observed in embryonic mortality or the incidence of chick abnormalities. However, TM significantly shortened hatch time by approximately 8.6 h compared to the control group (488.3 ± 4.2 h vs. 496.9 ± 5.3 h, *p* < 0.05). (Table 3).

As shown in Table 4, no significant differences in body weight (BW) were observed between the TM and control groups on day 7 post-hatch (*p* > 0.05). From day 14 to day 35, the control group exhibited significantly higher BW, average daily gain (ADG), and feed intake (FI) than the TM group (*p* < 0.05). However, the TM improved overall feed conversion ratio (FCR) across the 35-day period (1.40 ± 0.02 in TM vs. 1.51 ± 0.02 in control, *p* < 0.05).

Figure 2 depicts the effect of TM on body temperature (Tb) throughout the post-hatch period. TM did not result in significant changes in Tb compared to the control group at any time point post-hatch (*p* > 0.05).

On post-hatch day 35, the TM group exhibited notably reduced relative weights of the digestive system, heart, gizzard, proventriculus, and small intestine compared to the control group (Table 5). No differences were noted between the TM and control groups in terms of the length of the large intestine or the relative weights of the gallbladder, spleen, and liver.

### 3.2. Impact of TM and LPS Injection on Body Temperature (Tb) in Broiler Chickens

To assess the physiological response to inflammatory challenge, broiler chickens from TM and control incubation groups were intraperitoneally injected on day 15 with either lipopolysaccharide (LPS; 0.5 mg/kg BW) or sterile saline, and Tb and BW were recorded at 1, 3, and 6 h post-injection.

Tb and BW were measured individually for each bird (*n* = 10 per group). One-hour post-injection, LPS significantly reduced Tb in the control group (*p* < 0.05), indicating a hypothermic response, whereas no such change was observed in the TM group (Table 6). BW did not significantly differ among groups at 1, 3, or 6 h after LPS or saline injection (*p* > 0.05).

### 3.3. The Impact of TM and LPS Challenges on the mRNA Levels of Proinflammatory Cytokines (TNF-α, IL-1β, IL-2, IL-6, IL-12, and IFN-γ)

After one hour of the LPS challenge, both TM and control groups exhibited an increased mRNA expression of TNF-α, IL-1β, IL-2, IL-6, IL-12, and IFN-γ (Figure 3 and Figure 4) compared with corresponding saline groups. The control group showed higher IL-1β, IL-2, IL-12, and TNF-α levels, while IL-6 was elevated in the TM group (*p* < 0.05). At 3 h, IL-2 was elevated only in the TM group, with generally reduced expression in both groups. At this juncture, the TM group exhibited elevated levels of IL-1β, IL-2, IL-12, and TNF-α, whereas the expression of IL-6 was lower than that observed in the control group. By 6 h, IL-6 and IL-12 were significantly higher in TM, while IFN-γ and IL-2 remained higher in the control group. TNF-α returned to baseline in TM but remained elevated in the control group (*p* < 0.05).

### 3.4. Impact of TM and LPS Challenge on mRNA Levels of Anti-Inflammatory Cytokines (IL-10 and TGF-β)

At one-hour post-LPS injection, both IL-10 and TGF-β were upregulated in LPS challenged groups (Figure 5). The control group had significantly higher IL-10, while TM showed greater TGF-β expression. At three hours, IL-10 levels declined in TM but remained higher in the control. At six hours, both IL-10 and TGF-β were elevated in TM relative to the control (*p* < 0.05).

### 3.5. Impact of TM and LPS Challenge on the mRNA Levels of NF-κB and NF-κBp50 Transcription Factor

NF-κB expression increased one-hour post-LPS in both groups, with higher levels in TM (Figure 6). At three hours, expression declined, but TM maintained elevated levels compared to the control. At six hours, NF-κB decreased in TM and rose in the control. In contrast, NF-κBp50 was downregulated in the control group at 1 and 3 h but remained elevated in TM. At six hours, NF-κBp50 increased in the control but declined in TM (*p* < 0.05).

### 3.6. Impact of TM and LPS Challenge on Toll-like Receptors (TLRs) mRNA Levels (TLR 1, TLR 2, TLR 3, and TLR 4)

LPS injection significantly upregulated TLR-1 to TLR-4 mRNA expression at one hour in both challenged groups (Figure 7). The control group showed higher TLR-1 and TLR-3 expression, while TM showed increased TLR-2 and TLR-4. At three hours, all TLR expressions declined. TM maintained higher TLR-2 and TLR-4 than the control (*p* < 0.05). At six hours, TLR-2 decreased in TM but increased in the control; TLR-3 was elevated in both groups.

### 3.7. Impact of TM and LPS Challenge on mRNA Levels of Heat Shock Proteins and Heat Shock Factors (HSP70, HSP90, HSF-1, and HSF-3)

One hour after LPS injection, a sharp early peak in the expression of HSP70, HSF-1, and HSF-3 mRNA were observed in both the TM and control groups, accompanied by a decline in HSP90 expression in the TM group (Figure 8). At this point, TM showed reduced HSP70, HSP90, and HSF-1 expression but elevated HSF-3 compared to the control (*p* < 0.05). At three hours, HSP70 and HSP90 increased in TM, while HSF-1 and HSF-3 declined. At six hours, HSP70 and HSF-3 declined further in TM, while HSP90 and HSF-1 stabilized.

## 4. Discussion

Thermal manipulation involves altering temperature and humidity during avian egg incubation, implemented at cyclical intervals and designated periods throughout embryogenesis [15]. This method is cost-effective and straightforward to execute [16]. Additionally, TM was linked to changes in the basal and dynamic expression levels of key signaling proteins essential for preserving tissue integrity and supporting repair mechanisms [17]. This contributes to improved thermotolerance and immune response under heat stress [18]. Multiple studies have shown the positive effects of TM on Tb, BW, and immune system responses in post-hatch broiler chickens [15,19].

To assess whether embryonic TM influenced early developmental outcomes, we first examined hatching parameters, including hatchability and hatch time. In our work TM had no appreciable effect on the hatchability rate, in line with past research results [20,21]. Still, the impact of TM on hatchability rates could differ for several reasons. While some studies have shown higher hatchability rates following TM [22], others have noted decreased hatchability rates [23]. Variations in the genetic composition, age, and particular environmental conditions experienced during incubation—including elements like RH, temperature control, and pre-incubation background of the eggs—can help to explain this diversity.

Furthermore, the length and strength of the TM process greatly affect hatchability results. For example, intermittent (3–18 h/d) high (38.5–39.5 °C) TM between E07 and E18 can help offset the negative impacts of TM on hatchability and chick quality [24]. On the other hand, constant TM or incubation over 39.5 °C can drastically reduce hatchability, raise embryo mortality, and postpone hatching time, thereby affecting body weight by inadequate yolk sac absorption [25].

In this study, the hatch time for the TM group was 8 h earlier than that of the control group, supporting previous research findings [26]. Even a slight improvement in hatchability can bring considerable financial benefits to the commercial broiler industry; reducing the incubation period can significantly decrease production costs and time [25].

In this study, TM did not significantly impact the Tb of broiler chickens. Similarly, there was no notable difference in body temperature between the TM and control groups [27], and a comparable lack of effect was noted in ostrich chicks [28]. However, TM has been shown to change the levels of key thermoregulatory hormones, including plasma corticosterone, triiodothyronine, and testosterone, particularly during stress, possibly by inducing hypothalamic epigenetic modifications [29,30]. These hormonal changes might influence the observed effects of TM on thermoregulation by impacting metabolic rate [31,32,33,34]. Several studies suggest that TM reduces post-hatch body temperature and enhances the development of thermotolerance [8,9].

Following hatch, we evaluated growth performance metrics to determine whether TM conferred advantages in feed utilization or BW gain under standard rearing conditions. Compared to the control group in this study, the lower BW observed in the TM group aligns with previous findings [35], indicating that elevated incubation temperatures significantly reduce BW compared to normal incubation temperatures. Additionally, our results corroborate the findings of Piestun, et al. [36], who demonstrated that TM during embryogenesis decreased overall body weight (BW) and feed intake in broilers. However, feed efficiency and breast muscle mass increased in both sexes from hatching to 70 days of age.

Furthermore, numerous studies have demonstrated the association between post-hatching body weight and the TM protocol, emphasizing the necessity for optimization [8,37]. However, the significant decrease in BW of the Indian River breed in our study indicates that further adjustments to the TM procedure are needed for this breed. Additionally, different chicken breeds may respond differently to TM. For example, TM in layer-type chickens has yielded fewer advantageous outcomes compared to broilers [38].

TM broiler chickens used in this study showed reduced feed consumption and a more effective feed conversion ratio. This could be connected to better thermoregulation and a lower metabolic rate before and after hatching, thereby lowering the energy maintenance needs [39]. Furthermore, linked with lower feed intake, the TM group showed improved feed efficiency [36]. The results underline the need for temperature control techniques in maximizing broiler operating effectiveness and growth. This emphasizes the possible benefits of applying such methods in the sector of chicken raising.

Consistent with our results, reports of heart weight declining as incubation temperature rises have come from Christensen, et al. [40]. Either a decrease in the development of cardiac cells or an increase in the frequency of metabolic diseases connected to the evolution of the cardiovascular system may be the reason for this drop in heart weight and size [41]. Unlike the results of this investigation, Iraqi, et al. [30] claimed that TM had no appreciable effect on heart weight in recently hatched broiler chicks. Furthermore, consistent with Al-Zghoul and El-Bahr’s [42] earlier studies, the spleen weight in the current study showed no notable differences between the control and TM groups.

Unlike previous findings by Elsayed, et al. [43], which reported a significant reduction in liver weight percentage following increased incubation temperatures, our study did not observe a significant change in liver weight. However, TM significantly reduced the intestinal length, relative gizzard weight, and digestive system mass, aligning with previous reports that TM affects the gastrointestinal tract morphology [42]. These morphological changes may reflect an adaptive optimization of gut function, as TM has been linked to enhanced digestive enzyme activity, improved villus architecture, and greater nutrient absorption efficiency [18]. A more compact and functionally optimized intestine may reduce maintenance energy costs, allowing more energy to be diverted toward growth.

Moreover, TM may influence the early developmental programming of the gut microbiota, potentially enhancing microbiota–host interactions [44]. This improved symbiosis could contribute to a more efficient FCR, even in the presence of reduced intestinal morphological parameters.

The Gram-negative bacterial endotoxin LPS mimics a bacterial infection by promoting systemic inflammation and the release of proinflammatory cytokines [45]. The body employs this reaction to protect against infections, particularly those caused by Gram-negative bacteria. An LPS challenge may elicit a comparable response, albeit not precisely mirroring a bacterial challenge. This reaction is typically less intense and severe than a response to a bacterial infection [46].

We analyzed Tb responses following LPS injection to evaluate the acute physiological response to inflammatory challenge. Compared to the other groups, the control group that was administered LPS demonstrated a substantially lower body temperature one hour after the injection, indicating the onset of a hypothermic phase. De Boever, et al. [47] have characterized the early immunological response to LPS as a hypothermic phase, which is characterized by a decrease in blood pressure, an increase in deceased leukocytes, and a decrease in circulating white blood cells. Conversely, the hypothermic phase appears to be bypassed by the group administered TM LPS, implying that adaptability and resilience were improved. TM-induced changes in cytokine production or other immunomodulatory factors may explicate these responses [7,48].

To understand the molecular mechanisms underlying the LPS responses, we examined the temporal expression patterns of key inflammatory cytokines and heat shock response in the spleen at 1, 3, and 6 h post-injection.

In poultry, the expression of proinflammatory cytokines often increases in response to infections, indicating an activated immune response aimed at eliminating pathogens [49]. Prior research indicates that LPS enhances the production of cytokines, such as TNF-α, IL-1β, IL-2, IL-6, and IFN-γ, during the initial phase of inflammation [50].

After the LPS exposure, the initial cytokine identified in mammals is TNF-α, a crucial modulator of the inflammatory response. Plasma levels of TNF-α peak shortly following LPS administration [51]. TNF-α prompts monocytes or macrophages to secrete various proinflammatory cytokines, including IL-1β and IL-6, via feedback regulation [52]. In septic conditions, TNF-α generally manifests within 1.5 to 2 h post-LPS injection, succeeded by IL-1β and IL-6 [51]. The mRNA levels of TNF-α, IL-1β, and IL-6 in the spleen reached their maximum 1 h post-LPS injection in both groups during this study. The results differ from those reported by of Jing, et al. [50], which demonstrated that TNF-α peaked 2 to 3 h after LPS injection in layers. Their study’s previously observed peak levels of IL-1β and IL-6 align with our findings.

Relative to the control group, the TM group receiving saline shows higher expression of TNF-α, suggesting TM may boost the response of the immune system to modest tissue damage. One intraperitoneal saline injection has been demonstrated to cause a systemic stress response and raise TNF-α mRNA levels in the mouse frontal cortex [53]. This result shows that despite differences in the afflicted tissues, even minor physiological stimulation in the TM group can set off an inflammatory reaction.

Secreted by naïve T helper cells, IL-2 is a growth factor that drives T-cell proliferation [54]. It is crucial in several immunological processes, including defenses to bacterial infections [54]. Our results revealed dynamic changes in IL-2 mRNA expression; peak levels in both groups occurred six hours after LPS injection. This result runs counter to Yang, et al.’s [55] observation that repeated LPS injections had no effect on IL-2 mRNA expression in the broiler spleen. In layers within the first two hours after injection, Jing, et al. [50] noted dynamic fluctuations in IL-2 and IFN-γ levels. Three hours following injection, we observed a drop in IL-2 activity that fits the results of Gehad, et al. [56]. Lauw Fanny, et al. [57] nonetheless observed a decrease in IL-2 expression in vitro at 3 and 6 h following LPS administration.

The TM group’s IL-2 transcriptional activity decreased after 1 h and then peaked at 6 h post-injection. This time fluctuation implies a flexible immune response to the LPS challenge. The conflicting results indicate that experimental circumstances, including immunological stimulation and time of measurement, influence IL-2 expression kinetics. Further investigation is necessary to elucidate these variations and the impact of immunological challenges on IL-2 dynamics.

IL-6 significantly influences the immune response to bacterial infections. Dalrymple, et al. [58] reported the significance of IL-6 in the induction of neutrophils, a critical mechanism for the defense against infections. Furthermore, IL-6 demonstrates protective effects against LPS-induced endotoxemia by increasing the production of IL-10 and modulating cytokine responses, which includes the enhancement of hepatic antioxidant enzymes [59]. TM elevated the IL-6 gene expression compared to the control at 1 and 6 h post-injection in our study. This increase in IL-6 levels may suggest a more robust and protective immune response to Gram-negative bacteria and LPS challenge. In addition to promoting neutrophilia and improving the clearance of bacterial pathogens, TM could also mitigate the detrimental effects of LPS-induced inflammation by modulating antioxidant and anti-inflammatory pathways by increasing IL-6 production.

Furthermore, Shanmugasundaram, et al. [10] established that IL-6 expression levels in Pekin ducklings can be affected by alterations in TM procedures. A notable elevation in IL-6 expression was detected under high TM circumstances from embryonic day 0 to 25. This discovery aligns with the findings of our experiment conducted at 1 and 6 h post-LPS treatment. Our research demonstrates that TM decreases IL-6 expression at the 3 h mark, suggesting its impact on the temporal control of IL-6. The temporal regulation of IL-6 expression by TM may promote a more balanced immune response to the LPS assault. TM may enhance essential defensive mechanisms, including neutrophilia and hepatic antioxidant activity, by elevating IL-6 levels at both the early and late stages of the immune response, as articulated by Dalrymple, et al. [58] and Nandi, et al. [59]. Conversely, decreasing IL-6 in the TM group relative to the control group at 3 h may sustain the immune response and mitigate tissue damage, thus averting excessive inflammation.

Cell-mediated immunity is modulated by IL-12, which stimulates T cells and natural killer (NK) cells to generate interferon-gamma (IFN-γ) [60]. IL-12 has demonstrated considerable potential as a vaccine adjuvant to augment humoral immunity through its capacity to activate B cells [61]. Suppressing IL-12 in antigen-presenting cells is a critical mechanism that allows parasites to evade the immune system [62]. Our study demonstrated that LPS elevated IL-12 expression at both 1 and 6 h post-injection across both groups. TM markedly increased IL-12 expression at 3 and 6 h compared to the control group. The noted elevation of IL-12 in the TM group signifies enhanced activation of T cells and NK cells, leading to a more vigorous cell-mediated immune response. The results suggest that TM may enhance the immune system’s response to infections.

Interferon-gamma (IFN-γ) is a crucial cytokine in the immune response of broilers [63]. IFN-γ exhibits antiviral characteristics and is believed to mitigate heat stress in broilers via modulating the expression of heat shock protein 70 [64]. Our analysis revealed that IFN-γ mRNA expression exhibited a dynamic pattern in both groups following LPS injection, showing a significant increase one hour post-injection, a subsequent decline at three hours, and a peak at six hours post-injection. This trend aligns with the observations of Leshchinsky and Klasing [65], who noted an initial rise in IFN-γ at 1 h, succeeded by a decrease at 3 h in broilers. However, their study did not assess expression levels at the 6 h interval. In contrast, Jing, et al. [50] suggested that the initial post-injection phase is the optimal period for assessing IFN-γ expression and observed no significant changes at 6 h post-injection in the layers. The variations in IFN-γ expression patterns may be attributed to breed-specific immune response differences, since broilers and layers often exhibit distinct immunological profiles or alterations in experimental conditions, such as LPS dosage, timing, or sampling techniques.

Our study revealed that the control group exhibited a more pronounced early and late proinflammatory response than the TM group, underscoring the potential of TM in mitigating excessive inflammatory responses during immunological challenges. TM significantly reduced IFN-γ expression, consistent with Al-Zghoul’s [66] findings, which indicated decreased IFN-γ expression after heat stress in TM-exposed broilers. IFN-γ is a crucial cytokine that activates macrophages, mediates inflammation, and enhances Th1 immune responses. Its decreased levels in the TM group indicate a more regulated inflammatory response [67].

TM has been demonstrated to augment immunological responses in broiler chickens and is linked to the overexpression of cytokines that facilitate tissue integrity and repair during acute and chronic heat stress [68]. Moreover, TM can improve oxidative stress tolerance and modulate the expression levels of various splenic mRNA cytokines (IL-1β, IL-4, IL-6, IL-8, IL-15–18, IFN-α, IFN-β, IFN-γ, and TNF-α) in broiler chicken embryos, as well as several other critical genes implicated in immune cytokine induction pathways [69].

Anti-inflammatory cytokines regulate immune responses by preventing excessive inflammation in broiler chickens. For instance, IL-10 has been demonstrated to inhibit IFN-γ production in vitro [70]. In mammals, IL-10 inhibits proinflammatory cytokines such as TNF-α, IL-1β, IL-2, and IL-6, and it also affects the Th1 cell function by preventing IFN-γ synthesis. However, the role of IL-10 in poultry is still not well understood [70]. In our study, LPS injections induced IL-10 production in both groups, which aligns with findings from Jing, et al. [50] and Wang, et al. [71]. The TM group exhibited lower IL-10 levels at 1 and 3 h after LPS injection than the control group but showed higher levels at 6 h. This trend may be advantageous for TM, as although IL-10 supports immunological homeostasis, excessive IL-10 production can be leveraged by infections to undermine host defenses [72].

The TGF-β superfamily members are multifunctional signaling proteins that play pivotal roles in tissue development, homeostasis, and immune regulation [73]. TGF-β isoforms display a broad spectrum of biological activities, including regulating extracellular matrix production, controlling cell growth and differentiation, and suppressing immune activity due to their function as immune-suppressive cytokines [74]. Previously, it has been demonstrated that intraperitoneal lipopolysaccharide (IP-LPS) injection induces TGF-β expression in mice through TNF-α-mediated proteolytic activation of latent TGF-β [75]. This finding aligns with our study, which revealed that LPS challenge increased TGF-β expression levels.

Our results further indicate that TM modifies the dynamics of anti-inflammatory cytokine expression in response to an LPS challenge. Specifically, the TM group demonstrated a more potent and sustained expression of TGF-β, along with increased levels of IL-10, particularly at six hours post-LPS injection. This pattern may suggest a shift toward a regulatory and anti-inflammatory immune profile, which could aid in reducing excessive inflammation and its associated tissue damage.

Traditionally, NF-κB is a key transcriptional activator that regulates various proinflammatory, pro-survival, and pro-proliferative genes [76]. It is well established that most proinflammatory cytokine and chemokine genes contain NF-κB-binding sites in their promoter regions [77]. NF-κB activation is essential for triggering these genes in response to various immunostimulatory signals, including LPS [77]. However, NF-κB also demonstrates anti-inflammatory properties. For example, it can reduce the production of critical proinflammatory cytokines such as IL-1β, thereby regulating the inflammatory response [78].

LPS triggers the rapid degradation of IκBs—a family of inhibitory proteins that bind to NF-κB and prevent its nuclear localization signal—within minutes [79]. This process releases NF-κB, enabling its translocation to the nucleus regulating gene transcription [79]. In our study, the expression of the NF-κB gene was significantly upregulated after LPS treatment in both groups. This finding aligns with a previous study by Ansari, et al. [80], who reported increased NF-κB expression in the bursa at 12 and 36 h after IP-LPS injection. Likewise, increased NF-κB levels in the ileum were detected one day after the IP-LPS injection [81]. In accordance with our findings, Li, et al. [82] demonstrated that LPS enhances NF-κB expression in the broiler spleen following IP-LPS administration. These studies collectively corroborate our discovery of NF-κB activation in response to LPS.

In its active DNA-binding conformation, NF-κB assembles diverse dimers of distinct protein combinations from the NF-κB/Rel family. Among the five identified mammalian Rel proteins, NF-κB1 (p50 and its precursor p105) is a key example [83]. The p50 subunit of NF-κB is particularly important for its role in suppressing immune and inflammatory responses. It enables transcriptional repression via the canonical NF-κB pathway, which primarily controls inflammation [84].

Additionally, p50 can reduce the efficiency of NF-κB nuclear translocation and obstruct DNA binding, further amplifying its anti-inflammatory effects [85]. In our study, TM increased the expression of NF-κB p50 and NF-κB at 1 and 3 h after LPS injection, compared to the control group. This implies that TM may improve the regulatory mechanisms of NF-κB, potentially promoting a balanced inflammatory response and restricting excessive inflammation. The specific effects of LPS and TM on the NF-κB p50 and NF-κB signaling pathways necessitate additional research to elucidate their roles in regulating immunological and inflammatory responses.

Essential for organizing the host’s immune response to different pathogenic organisms, TLRs are recognized as a main class of pattern recognition receptors (PRRs) [86]. Recognizing pathogen-associated molecular patterns (PAMRs), pattern recognition receptors (PRRs) set off signal transduction pathways that result in later immune activation [87]. So far, several functioning TLRs have been found in chickens [88]. Among these, TLR-4 is considered the primary receptor for LPS recognition [87]. However, other TLRs may also interact and cooperate in response to LPS stimulation [89].

LPS injection elevated the mRNA expression of TLR-1, TLR-2, TLR-3, and TLR-4 in both groups. The findings correspond with those of Srivastava, et al. [90], who demonstrated that LPS stimulation in human cell cultures increased the expression of TLR1, TLR2, TLR3, and TLR4 within one hour. They found that TLR1 levels increased six hours later, whereas TLR2, TLR3, and TLR4 levels decreased. In contrast, our work revealed that while TLR3 levels were elevated, LPS reduced the expression of TLR1, TLR2, and TLR4 after six hours. Discrepancies may result from species-specific responses to LPS or from the use of varying experimental configurations, such as cell cultures instead of live animals.

Moreover, our investigation showed that following injection, LPS first raised TLR-4 expression in both groups then caused a decrease. This result is in line with past research showing higher broiler TLR-4 gene expression in vivo following IP LPS injection [91]. Eren, et al. [92] found opposing findings, though, whereby following IP-LPS administration, TLR-2 and TLR-4 intensity in broiler cecal tissue showed no appreciable change. They linked this disparity to either a low LPS dosage [92] or underdevelopment of innate immunity during the experimental period. These results imply that LPS starts a dynamic, time-dependent modification of TLR expression that might differ based on the tissue type, LPS dosage, and developmental stage of the birds.

Our research indicated that TM modulates the immunological stress response via its varied impacts on TLR expression. TM specifically modified TLR expression patterns in response to LPS challenge. For example, TM elevated the expression levels of TLR-2 and TLR-4 following LPS administration. Overexpression may enhance the capacity of TM-treated avians to react to Gram-negative bacteria via TLR4, identified as the principal receptor for LPS recognition [87]. Six hours post-LPS injection, the TM group exhibited a reduced overall TLR expression, apart from TLR1. This result further corroborates the observations of diminished proinflammatory and elevated anti-inflammatory cytokine levels. Prior studies indicate that heat stress diminishes TLR-4 expression in chicks subjected to TM therapy [93]. The many processes associated with each challenge can elucidate the observed disparity—LPS versus heat stress. The differential control of TLR expression mediated by TM highlights its potential role in maintaining immunological homeostasis during inflammatory circumstances.

After facing proteotoxic stresses like heat shock, cold, oxidative stress, hypoxia, toxins, chemicals, and infections, the heat shock response (HSR) is a vital adaptive mechanism helping in restoring cellular homeostasis [94]. Frequently causing protein denaturation and misfolding, cellular stress either causes cellular damage or even cell death [95]. By means of the HSR, molecular chaperones—including heat shock proteins (HSPs) and heat shock factors (HSFs)—that control HSP expression counteract these effects [96]. Together, these elements preserve the integrity of structural proteins, stop protein aggregation, and help damaged proteins be sufficiently folded and refolded [97].

HSP70 plays a critical role in the cellular response to bacterial infections and inflammation induced by lipopolysaccharides. Hsp70 provides protection against inflammation caused by both Gram-negative and Gram-positive bacteria during bacterial infections. It reduces inflammatory responses induced by lipopolysaccharide (LPS) and lipoteichoic acid (LTA) [98]. The observed increase in response to LPS in our study aligns with its role in mitigating cellular stress associated with inflammatory reactions. Critical immunological signaling pathways, such as NF-κB, utilized by the host to combat bacterial infections, also exhibit interactions with HSP70 [99]. The elevated expression of HSP70 observed in our study may be associated with its role in enhancing the immune response to LPS exposure.

In our work, however, HSP70 expression was lowered at 1 and 6 h post-injection in the TM group as compared to the control group. This drop in HSP70 levels in the TM group could be connected to the noted early- and late-phase immune response-related decrease in general inflammatory cytokines, including TNF-α, IL-1β, IFN-γ, and IL-2. Human monocytes, including TNF-α, IL-1β, and IL-6, have shown HSP70 to upregulate the gene expression of proinflammatory cytokines [99]. Thus, the lower levels of HSP70 in the TM group could help explain the decreased expression of these cytokines by implying that TM may control the inflammatory response by adjusting HSP70 expression.

Based on the non-significant changes in the control group, our results show that LPS injections had a minimum impact on HSP90 gene expression. Still, LPS had a significant impact on HSF-1, HSF-3, and HSP70 gene expression. These findings contrast with those of Miller and Qureshi [100] who found that in chicken monocytic cells, in vitro, LPS stimulation raised HSP70 and HSP90 expression. In mouse cell cultures under LPS stimulation, similarly higher expression of both genes was noted. Fascinatingly, three hours post-LPS injection, the expression of HSP70 and HSP90 in the TM group corresponds with these results, implying that TM might improve the immune response at the cellular and molecular levels [101]. More studies are required to understand how LPS completely influences heat shock response.

Post-LPS challenges the immunological response, which is regulated significantly by HSFs. Chen, et al. [102] recorded a more severe immunological response, which resulted in several organ malfunctions and a reduced survival rate in HSF-1 knockout mice. They also detailed how HSF-1 controls the generation of inflammatory cytokines. Since they are the two main HSFs that react to heat stress in avian species, HSF-1 and HSF-3 are considered avian-specific HSFs [103]. In our work, LPS injections raise HSFs in broiler spleens, emphasizing their relevance in controlling the cellular stress response during inflammation.

TM decreased HSF-1 expression at 1 and 6 h post-injection compared to the control. HSF1, a principal transcriptional regulator of HSP70, modulates its expression by releasing RNA polymerase II (Pol II) from promoter-proximal pausing [104]. HSF-1 is associated with heat shock elements (HSEs) in the HSP70 promoter during stress conditions, thereby recruiting co-activators (e.g., P-TEFb) that phosphorylate RNA Pol II to liberate it from a paused state, thus promoting transcriptional elongation and enhanced HSP70 production [104]. The TM group’s observed reduction in HSF-1 expression undoubtedly disrupted this critical regulatory period. Decreased HSF-1 levels hinder the recruitment of transcriptional machinery to the HSP70 promoter, resulting in inadequate release of stalled RNA Pol II and thus diminished HSP70 production. This method may elucidate the reduced HSP70 levels in the TM group at 1 and 6 h post-injection.

Our findings indicate that TM elevated the expression of HSF-3 one hour after LPS injection, while a decrease in its expression was observed three hours post-injection compared to the control group. HSF-3 is essential for heat shock response and thermotolerance in birds. For example, the knockout of HSF3 in avian cells results in a significant decrease in heat shock gene expression and a loss of thermotolerance despite normal HSF1 levels [105]. This underscores the significant function of HSF-3 in avian stress responses and indicates that TM might influence the heat shock response by variably regulating HSF-3 expression.

Furthermore, the reduced level of HSP70 in the TM group one hour after LPS injection, followed by its increase at three hours compared to the control group, supports the hypothesis that Hsp70 may play a role in negatively regulating Hsf-3 activation. Al-Zghoul, et al. [106] observed that the downregulation of the Hsf-3 gene is simultaneously associated with the upregulation of the Hsp70 gene in the pectoral and thigh muscles and the brain. This inverse relationship suggests that when sufficient levels of HSP70 are produced to alleviate cellular stress, HSP70 may act as a feedback regulator, consequently suppressing HSF-3 activity.

Our findings underscore the potential role of TM in regulating the HSR, which is essential for cell survival through the processes of protein folding and unfolding in response to stressors such as LPS exposure. TM provides a quantifiable response that mitigates excessive inflammation while maintaining cellular homeostasis through the variable regulation of HSPs and HSF expressions. This regulatory system has the potential to enhance TM-treated birds’ resilience to stress, thereby improving immunological responses and overall health.

A schematic summary of these proposed mechanisms is illustrated in Figure 9, highlighting how TM enhances anti-inflammatory signaling, suppresses proinflammatory cytokines, upregulates TLR expression, and promotes a more adaptive heat shock response, ultimately reducing inflammatory burden and supporting resilience to immune stressors.

This study has several limitations that should be acknowledged. First, although saline-injected controls were used to reduce injection-related confounding, the absence of a completely non-injected (naïve) group limits the ability to fully define baseline gene expression levels. Second, while multiple cytokine and gene expression markers were analyzed across several time points, we did not apply a global multiple testing correction, such as the Benjamini–Hochberg procedure, which may increase the risk of Type I error. Third, although statistical significance was used to interpret treatment effects, effect sizes were not uniformly reported, which may limit the assessment of biological relevance. Lastly, the findings are limited to a controlled experimental setting, and the extent to which these immunomodulatory effects of thermal manipulation translate to commercial field conditions remains to be determined. Future research should address these points by including uninjected controls, applying formal false discovery rate adjustments, and expanding the scope to field-relevant trials. While TM protocols can be precisely controlled in experimental incubators, scaling this approach to industrial settings may require modifications to standard incubation equipment, enhanced monitoring systems, and staff training to ensure consistent temperature and humidity regulation. Additionally, economic feasibility, compatibility with large-scale hatching schedules, and potential variability across genetic lines are important considerations. These challenges must be addressed through pilot-scale trials and industry collaboration before routine adoption of TM can be recommended in commercial poultry production.

## 5. Conclusions

In summary, thermal manipulation (TM) during embryonic development improved the immune response of broiler chickens to a post-hatch lipopolysaccharide (LPS) challenge. While TM did not affect hatchability, it shortened hatch time and improved feed efficiency. TM also reduced excessive inflammation by regulating proinflammatory and anti-inflammatory cytokines, transcription factors, and toll-like receptors. Additionally, TM influenced the expression of heat shock proteins and their regulators, supporting better stress response. Overall, TM shows promise as a non-antibiotic strategy to enhance broiler health and resilience under stress.

## Figures and Tables

**Figure 1 animals-15-01736-f001:**
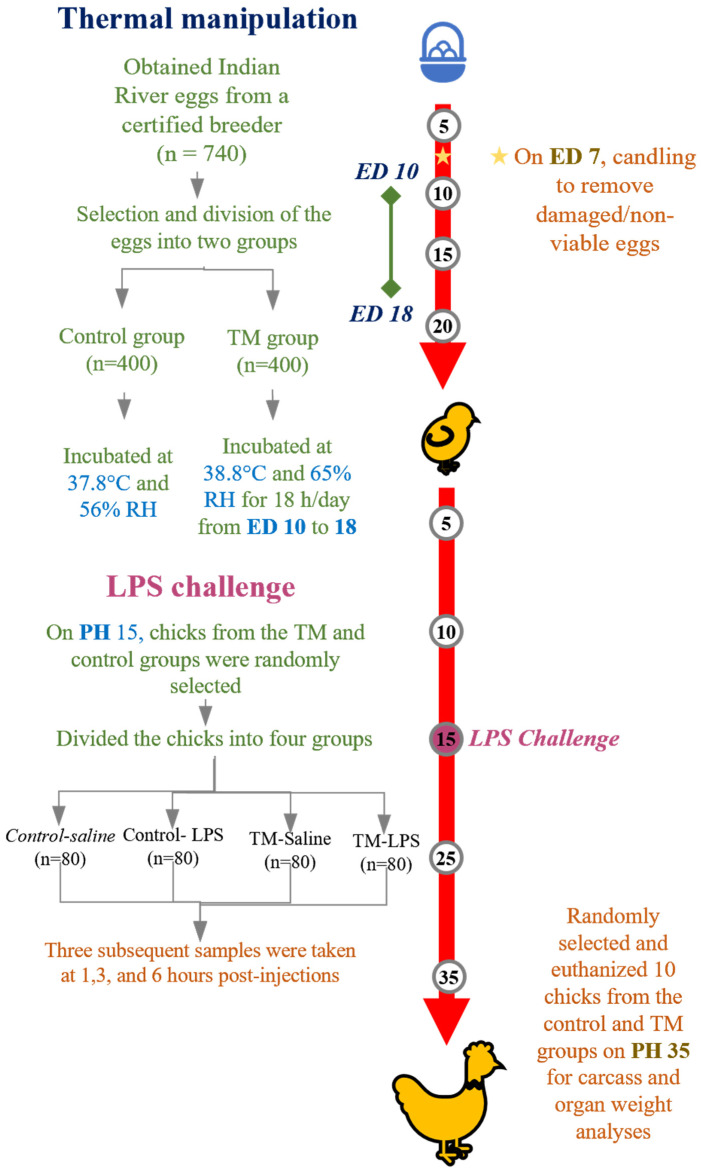
Summarizes the experimental setup used in this investigation. RH stands for relative humidity, LPS for lipopolysaccharide, ED for embryonic day, PH for post-hatch, TM for thermal manipulation.

**Figure 2 animals-15-01736-f002:**
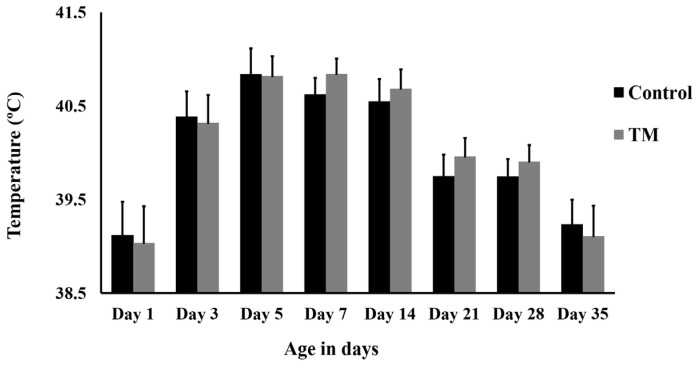
Thermal manipulation (TM) impact on body temperature (38.8 °C, 65% relative humidity (RH) 18 h daily during embryonic days 10–18).

**Figure 3 animals-15-01736-f003:**
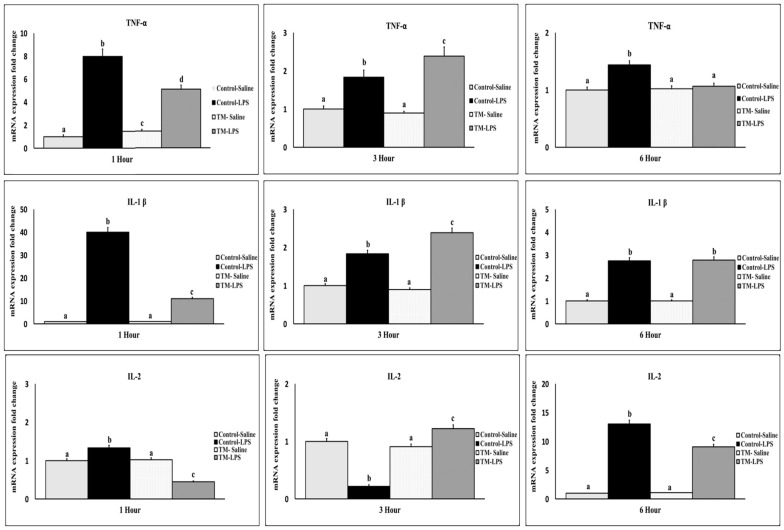
Impact of thermal manipulation (TM) at 38.8 °C and 65% relative humidity (RH) for 18 h daily during embryonic days (EDs) 10–18 on the splenic mRNA levels of proinflammatory cytokines (TNF-α, IL-1β, and IL-2) in broiler chickens after 1, 3 and 6 h post-LPS challenge. a–d: Within the Time interval 1, 3, and 6 h post-LPS challenge, the mean of the TM group is significantly different compared to the mean of the control (*p* < 0.05).

**Figure 4 animals-15-01736-f004:**
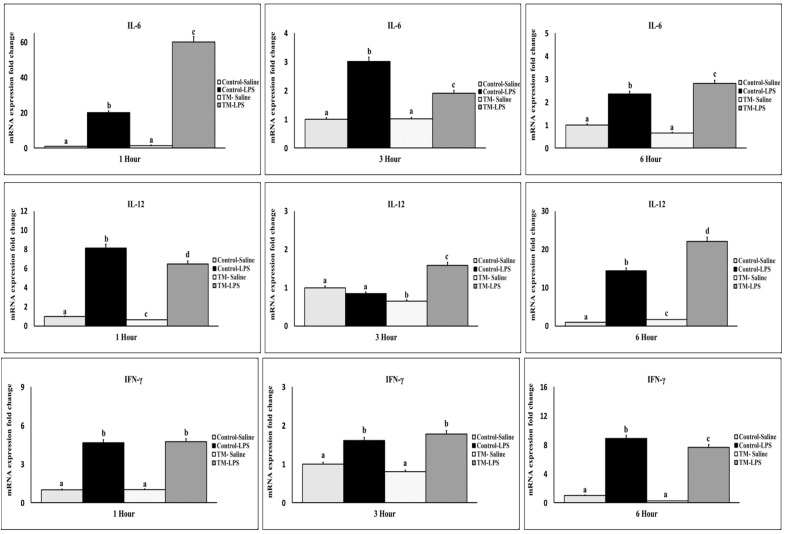
Impact of thermal manipulation (TM) at 38.8 °C and 65% relative humidity (RH) for 18 h daily during embryonic days (Eds) 10–18 on the splenic mRNA levels of proinflammatory cytokines (IL-6, IL-12, and IFN-γ) in broiler chickens after 1, 3 and 6 h post-LPS challenge. a–d: Within the Time interval 1, 3, and 6 h post-LPS challenge, the mean of the TM group is significantly different compared to the mean of the control (*p* < 0.05).

**Figure 5 animals-15-01736-f005:**
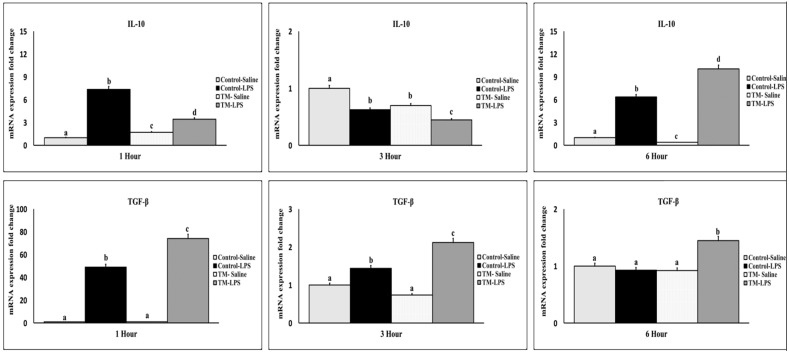
Impact of thermal manipulation (TM) at 38.8 °C and 65% relative humidity (RH) for 18 h daily during embryonic days (EDs) 10–18 on the splenic mRNA levels of anti-inflammatory cytokines (IL-10 and TGF-β) in broiler chickens after 1, 3 and 6 h post-LPS challenge. a–d: Within the Time interval 1, 3, and 6 h post-LPS challenge, the mean of the TM group is significantly different compared to the mean of the control (*p* < 0.05).

**Figure 6 animals-15-01736-f006:**
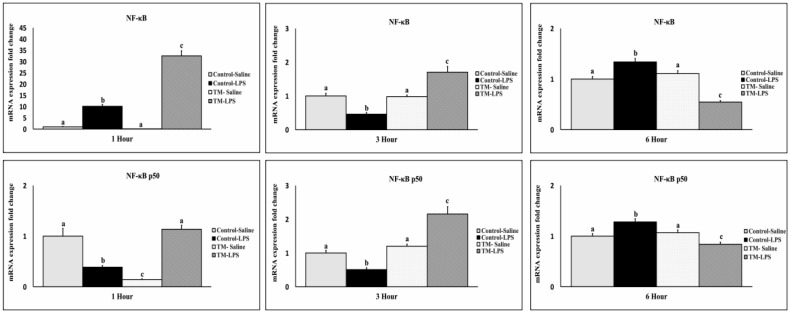
Impact of thermal manipulation (TM) at 38.8 °C and 65% RH relative humidity (RH) for 18 h daily during embryonic days (EDs) 10–18 on the splenic mRNA levels of NF-κB and NF-κBp50 in broiler chickens after 1, 3, and 6 h post-LPS challenge. a–c: Within the Time interval 1, 3, and 6 h post-LPS challenge, the mean of the TM group is significantly different compared to the mean of the control (*p* < 0.05).

**Figure 7 animals-15-01736-f007:**
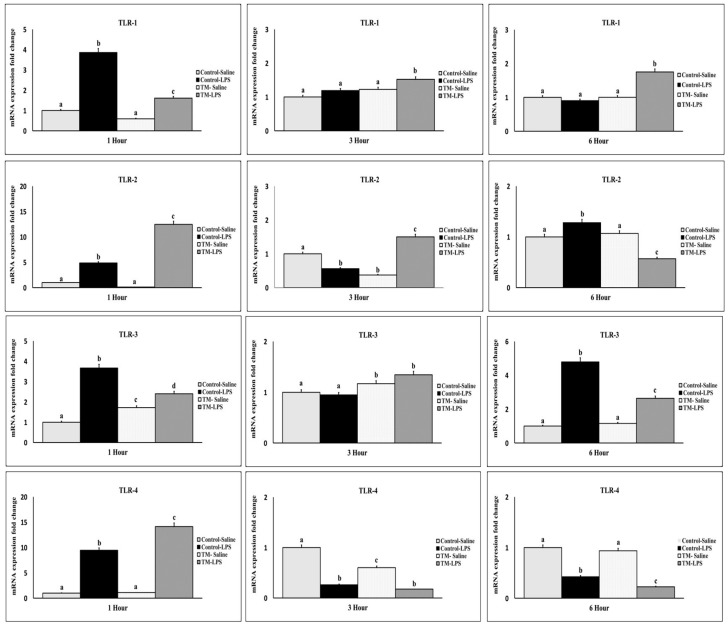
Impact of thermal manipulation (TM) at 38.8 °C and 65% relative humidity (RH) for 18 h daily during embryonic days (EDs) 10–18 on the splenic mRNA levels of toll-like receptor (TLR) mRNA levels (TLR 1, TLR 2, TLR 3, and TLR 4) in broiler chickens after 1, 3 and 6 h post-LPS challenge. a–d: Within the Time interval 1, 3, and 6 h post-LPS challenge, the mean of the TM group is significantly different compared to the mean of the control (*p* < 0.05).

**Figure 8 animals-15-01736-f008:**
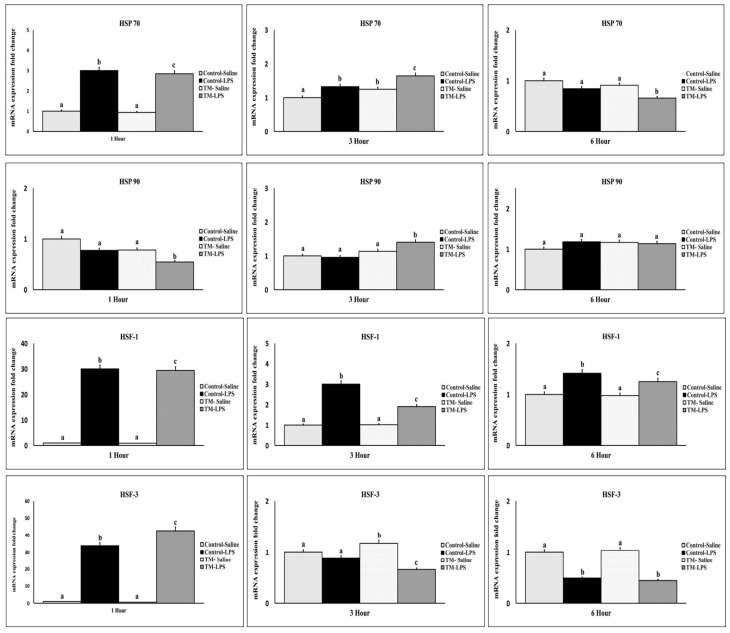
Impact of thermal manipulation (TM) at 38.8 °C and 65% relative humidity (RH) relative humidity (RH) for 18 h daily during embryonic days (EDs) 10–18 on the splenic mRNA levels of Heat Shock Proteins and Heat Shock Factors (HSP70, HSP90, HSF-1, and HSF-3) in broiler chickens after 1, 3 and 6 h post-LPS challenge. a–c: Within the Time interval 1, 3, and 6 h post-LPS challenge, the mean of the TM group is significantly different compared to the mean of the control (*p* < 0.05).

**Figure 9 animals-15-01736-f009:**
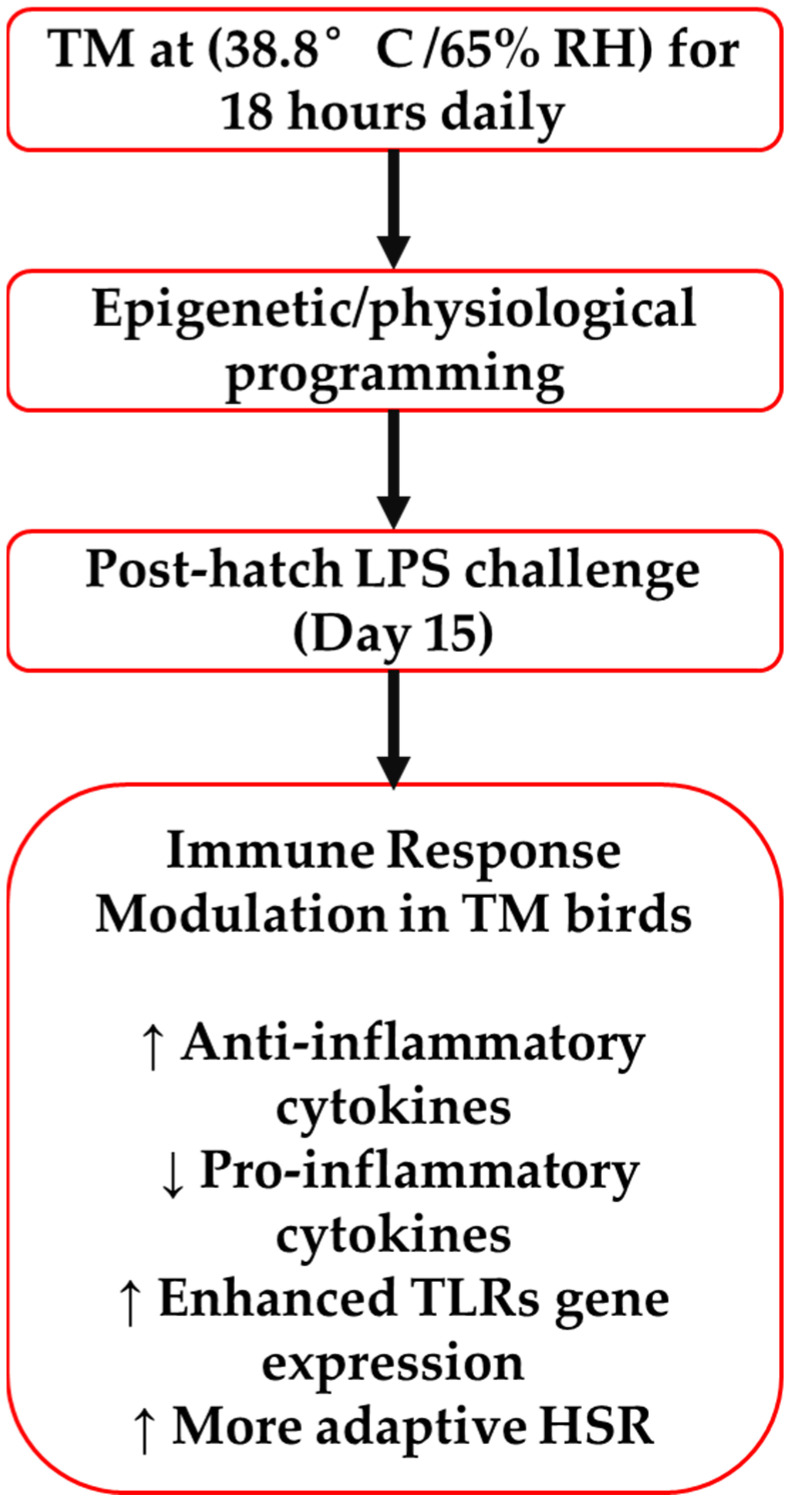
Proposed schematic model of immune modulation induced by embryonic thermal manipulation (TM) in broiler chickens. TM leads to increased anti-inflammatory cytokine expression, reduced proinflammatory cytokines, enhanced toll-like receptor (TLR) gene expression, and a more adaptive heat shock response (HSR), ultimately contributing to reduced inflammatory burden and improved resilience.

**Table 1 animals-15-01736-t001:** Ingredients and calculated chemical composition of basal experimental diets presented on an as-fed basis. Nutrient values were calculated according to NRC (1994) [14].

Ingredients (%)	Starter (d1–24)	Grower (d25–35)
Corn	60.20	62.0
Soybean Meal (CP 40%)	34.30	31.50
Concentrate	3.20	2.70
Vegetable Oil	1.00	2.50
Limestone	1.10	1.10
Vitamin and mineral premix *	0.20	0.20
Total	100.00	100.00
*Nutrient Composition* (*Calculated*) ****		
Metabolizable energy, kcal/kg	3110	3155
Crude Protein (%)	22.52	21.53
Lysine (%)	1.51	1.32
Methionine (%)	0.55	0.51
Calcium	1.08	1.0
Available Phosphorus	0.49	0.44

* Vitamin and Mineral premix, each 2 kg consists of the following: Vit A 12,000,000 IU; Vit D3, 2,000,000 IU; Vit. E. 10 g; Vit k3 2 g; Vit B1, 1000 mg; Vit B2, 49 g; Vit B6, 105 g; Vit B12, 10 mg; Pantothenic acid, 10 g; Niacin, 20 g, Folic acid, 1000 mg; Biotin, 50 g; Choline Chloride, 500 mg, Fe, 30 g; Mn, 40 g; Cu, 3 g; Co, 200 mg; Si, 100 mg and Zn, 45 g. ** Calculated according to NRC (1994) [14]. Note: No common salt (NaCl) was added separately, as the basal ingredients met the sodium requirements.

**Table 2 animals-15-01736-t002:** Primer sequences that are used in the real-time qPCR analysis.

The Gene	Sequence (5′–3′)	TM (°C)
β-Actin	F: ACCGCAAATGCTTCTAAACC R: ATAAAGCCATGCCAATCTCG	60
GAPDH	F: TTGTCTCCTGTGACTTCAATGGTG R: ACGGTTGCTGTATCCAAACTCAT	60
TNF-α	F: GACAGCCTATGCCAACAAGTA R: TTACAGGAAGGGCAACTCATC	60
IL-1 β	F: CCCGCCTTCCGCTACA R: CACGAAGCACTTCTGGTTGATG	60
IL-2	F: GAGAGCATCCGGATAGTGAAT R: TGTGGAGGCTTTGCATAAGAG	60
IL-6	F: AAATCCCTCCTCGCCAATCT R: CCCTCACGGTCTTCTCCATAAA	60
IL-12	F: CTGTGGCTCGCACTGATAAA R: CAATGACCTCCAGGAACATCTC	60
IFN-γ	F: ACCTTCCTGATGGCGTGAAG R: GCGCTGGATTCTCAAGTCGT	60
IL-10	F: CAGCAATCCAGAGACGATGAA R: AGTGGACTTGCACTGGAATAG	60
TGF-β	F: GGGTGTCCCATACCATTTAGAG R: CCCTTTAACGCAGAGGGATT	60
NF-κB	F: GATGTGGAGACAGACAGCTAAC R: CATAAGACGCACCACACTGA	60
NF-κB p50	F: CACAGCTGGAGGGAAGTAAAT R: TTGAGTAAGGAAGTGAGGTTGAG	60
TLR 1	F: AGTCCATCTTTGTGTTGTCGCC R: ATTGGCTCCAGCAAGATCAGG	60
TLR 2	F: GATTGTGGACAACATCATTGACTC R: AGAGCTGCTTTCAAGTTTTCCC	60
TLR 3	F: TCAGTACATTTGTAACACCCCGCC R: GGCGTCATAATCAAACACTCC	60
TLR 4	F: AGTCTGAAATTGCTGAGCTCAAAT R: GCGACGTTAAGCCATGGAAG	60
HSP 70	F: CGGGCAAGTTTGACCTAA R: TTGGCTCCCACCCTATCTCT	60
HSP 90	F: TCCTGTCCTGGCTTTAGTTT R: AGGTGGCATCTCCTCGGT	60
HSF 1	F: CAGGGAAGCAGTTGGTTCACTACACG R: CCTTGGGTTTGGGTTGCTCAGTC	60
HSF 3	F: GAGTTCCAGCACCCTTTCTT R: TCTTTCCACAGGGCCTTATTT	60

**Table 3 animals-15-01736-t003:** Effect of thermal manipulation (TM) at 38.8 °C and 65% RH for 18 h daily during embryonic days (EDs) 10–18 on hatching parameters (Control: 37.8 °C, 56% RH throughout).

Parameter	CON	TM	*p*-Value
True hatchability % ^1^	93.03 ± 0.541 ^a^	93.71 ± 1.092 ^a^	0.633
Unfertilized eggs % ^2^	4.36 ± 0.316 ^a^	1.72 ± 0.015 ^b^	0.014
Hatch time (h)	496.9 ± 5.3 ^a^	488.3 ± 4.2 ^b^	0.040
Embryonic death % ^3^	2.61 ± 0.857 ^a^	3.72 ± 0.828 ^a^	0.451
Hatchability of fertile % ^4^	97.27 ±0.888 ^a^	95.35 ± 1.126 ^a^	0.311
Chick Abnormalities % ^5^ (at hatching)	0.94 ± 0.313 ^a^	3.36 ± 1.519 ^a^	0.259

Values within a row with different superscripts differ significantly at *p* < 0.05. ^1^ Total eggs hatched/Total eggs incubated × 100%. ^2^ Unfertilized eggs/Total eggs incubated × 100%. ^3^ Embryonic death/Total eggs incubated × 100%. ^4^ Hatched eggs/Total fertile eggs × 100%. ^5^ Abnormal chick hatched/total eggs hatched × 100%.

**Table 4 animals-15-01736-t004:** Impact of thermal manipulation (TM) at 38.8 °C and 65% RH for 18 h daily during embryonic days (EDs) 10–18 on body weight (BW), Average Daily Feed Intake (ADFI), Average Daily Gain (ADG), and Feed Conversion Ratio (FCR) (Control: 37.8 °C, 56% RH throughout incubation).

Item	Group	Day 7	Day 14	Day 21	Day 28	Day 35	Overall (Day 1–35)
BW	CON	182 ± 2.13 ^a^	511.04 ± 4.07 ^a^	1126.25 ± 10.91 ^a^	1911.01 ± 18.64 ^a^	2729.29 ± 38.11 ^a^	2729.29 ± 38.11 ^a^
	TM	183.33 ± 1.4 ^a^	481.33 ± 5.60 ^b^	1048.75 ± 11.65 ^b^	1764.68 ± 15.71 ^b^	2519.51 ± 32.81 ^b^	2519.51 ± 32.81 ^b^
ADFI	CON	25.22 ± 0.18 ^a^	59.29 ± 1.14 ^a^	116.39 ± 1.37 ^a^	170.58 ± 1.56 ^a^	207.55 ± 2.70 ^a^	115.81 ± 0.88 ^a^
	TM	23.13 ± 0.26 ^b^	54.67 ± 1.23 ^b^	105.84 ± 1.46 ^b^	152.66 ± 1.92 ^b^	168.53 ± 3.89 ^b^	100.96 ± 0.90 ^b^
ADG	CON	20.34 ± 0.32 ^a^	47 ± 0.49 ^a^	87.89 ± 1.57 ^a^	112.11 ± 2.44 ^a^	116.83 ± 5.29 ^a^	76.85 ± 1.09 ^a^
	TM	20.52 ± 0.30 ^a^	42.57 ± 0.85 ^b^	81.06 ± 1.20 ^b^	102.28 ± 1.99 ^b^	107.49 ± 5.07 ^a^	70.85 ± 0.96 ^b^
FCR	CON	0.97 ± 0.01 ^a^	1.26 ± 0.03 ^a^	1.33 ± 0.02 ^a^	1.53 ± 0.03 ^a^	1.85 ± 0.12 ^a^	1.51 ± 0.02 ^a^
	TM	0.88 ± 0.01 ^b^	1.29 ± 0.03 ^a^	1.31 ± 0.02 ^a^	1.50 ± 0.03 ^a^	1.59 ± 0.08 ^a^	1.40 ± 0.02 ^b^

Values within a column with different superscripts differ significantly at *p* < 0.05. BW is expressed in grams (g), ADFI and ADG are expressed in grams per bird per day (g/bird/day), and FCR is a unitless value (g feed/g gain).

**Table 5 animals-15-01736-t005:** Impact of thermal manipulation (TM) at 38.8 °C and 65% RH for 18 h daily during embryonic days (EDs) 10–18 on internal organ weight (g) and small and large intestine length (mm) of Indian River male broilers on post-hatch day 35 (Control: 37.8 °C, 56% RH throughout incubation).

Organ/Tissue	CON	TM	*p*-Value
Liver			0.345
Absolute weight (g)	59.77 ± 2.87 ^a^	51.90 ± 2.52 ^a^	
Relative weight	2.19 ± 0.105 ^a^	2.06 ± 0.10 ^a^	
Spleen			0.561
Absolute weight	3.55 ± 0.38 ^a^	3.02 ± 0.23 ^a^	
Relative weight	0.13 ± 0.014 ^a^	0.12 ± 0.009 ^a^	
heart			0.001
Absolute weight	13.73 ± 0.55	9.47 ± 0.66	
Relative weight	0.503 ± 0.020 ^a^	0.376 ± 0.026 ^b^	
Small Intestine			<0.0001
Absolute weight	76.96 ± 5.05 ^a^	39.30 ± 2.60 ^b^	
Relative weight	2.82 ± 0.185 ^a^	1.56 ± 0.103 ^b^	
Large Intestine			0.053
Absolute weight	21.83 ± 2.46 ^a^	14.87 ± 1.01 ^a^	
Relative weight	0.80 ± 0.09 ^a^	0.59 ± 0.04 ^a^	
digestive System			<0.0001
Absolute weight	265.82 ± 9.83 ^a^	173.34 ± 11.34 ^a^	
Relative weight	9.74 ± 0.36 ^a^	6.88 ± 0.45 ^b^	
Gizzard weight			0.0042
Absolute weight	45.58 ± 1.45 ^a^	32.25 ± 2.67 ^b^	
Relative weight	1.67 ± 0.053 ^a^	1.28 ± 0.106 ^b^	
proventriculus			<0.001
Absolute weight	9.28 ± 0.49 ^a^	5.80 ± 0.43 ^b^	
Relative weight	0.34 ± 0.018 ^a^	0.23 ± 0.017 ^b^	
gallbladder			0.644
Absolute weight	0.79 ± 0.08 ^a^	0.78 ± 0.08 ^a^	
Relative weight	0.029 ± 0.003 ^a^	0.031 ± 0003 ^a^	
Small Intestine length (mm)	202.0 ± 7.66 ^a^	161.0 ± 5.39 ^b^	<0.001
Large intestine length (mm)	27.50 ± 1.73 ^a^	2.052 ± 6.21 ^a^	0.287

Values within a row with different superscripts differ significantly at *p* < 0.05. Relative weight presented as g/100 g body weight.

**Table 6 animals-15-01736-t006:** Impact of thermal manipulation (TM) at 38.8 °C and 65% RH for 18 h daily during embryonic days (EDs) 10–18 on body temperature (°C) at 1, 3, and 6 h after LPS challenge (Control: 37.8 °C, 56% RH throughout incubation).

Before LPS Injection	(°C)	CON-Saline	Control-LPS	TM-Saline	TM-LPS
		40.36 ± 0.32	40.55 ± 0.11 ^a^	40.79 ± 0.18	40.80 ± 0.22
1 h After LPS injection	(°C)	CON-Saline	Control-LPS	TM-Saline	TM-LPS
		40.19 ± 0.25	40.19 ± 0.14 ^b^	40.60 ± 0.15	40.34 ± 0.36
3 h After LPS injection	(°C)	CON-Saline	Control-LPS	TM-Saline	TM-LPS
		40.43 ± 0.18	40.33 ± 0.17	40.68 ± 0.20	40.86 ± 0.17
6 h After LPS injection	(°C)	CON-Saline	Control-LPS	TM-Saline	TM-LPS
		40.40 ± 0.27	40.59 ± 0.17	40.85 ± 0.29	40.61 ± 0.20

Values within a column with different superscripts differ significantly at *p* < 0.05.

## Data Availability

The original contributions presented in this study are included in the article. Further inquiries can be directed to the corresponding author.

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
