# Peer review of "Impact of Thermal Manipulation of Broiler Eggs on Growth Performance, Splenic Inflammatory Cytokine Levels, and Heat Shock Protein Responses to Post-Hatch Lipopolysaccharide (LPS) Challenge"

_animals, 2025, doi:10.3390/ani15121736_

Round 1

Reviewer 1 Report

Comments and Suggestions for Authors

The manuscript quality is inferior. There are numerous fundamental errors in this manuscript. Many mistakes and queries need to be corrected and responded.

Recommendations for revision

Generally: I recommend that a native English speaker read the manuscript.

Questions

Line 103: What is the average weight of fertile eggs?

Line 105: Change “These eggs were then divided ” to “Then these eggs were divided”.

Line 115 and Line 118: The font sizes are different.

Line 120: Change “each pen ” to “each cage”.

Line 268: the header “after 1, 3, and 6h.” is wrong, should be “at 1, 3, and 6h”

Line 107-109: What is the basis for choosing this condition? embryonic days (ED) 10 and 18, 38.8℃ with 65% RH, 18 hours each day? In addition, why are the humidity levels different between the two groups?

Line 118-119: “All chickens were feather-sexed with a male-to-female ratio of 47.3% to 52.7%. A total of 320 broiler males were evenly divided into two groups.”. the experiment design was wrong, in other words, A total of 160 chicks were selected per group (NC and TM).

Line 138: “The room temperature was maintained at 30±1 °C during the first three days...”,The environment temperature is low for the first three days. Generally, 1 d temperature: 34-35℃. In this study, the chick could be dead when environment temperature is 30±1 °C.

Line 144: This is matter of principle, please declare. The purpose of this research is explore whether the TM method could improved LPS-induced immune deficiency, but LPS was injected only at 15 days of age, why do not choose 35 days of age.

Line 210: A two-way ANOVA was utilized to evaluate data, but superscripts in Table 6 was “✳” rather than a,b. Meanwhile, Significant changes in body temperature was showed between CON-Saline and Control-LPS (40.55±0.11 VS 40.55±0.11), unacceptably.

Line 436-442: This study do not measure critical hormones, so, this paragraph should be deleted.

Line 449-451: The result is non-significant, but is significant in discussion. Why?

Materials and Methods

This experiment design should be divided into two, experiment 1 and experiment 2. Experiment 1 (Fertile eggs): NC VS TM. Experiment 2 (Hatching): NC+LPS VS TM+LPS.

Results

A large number of target gene expressions were detected, but some are  useless.

Discussion

Excessive content.

Tables

Table 1: Why the period is Starter (d1-14) and Grower (d25-35)? Where is the the period d15-24?

Table 1: Header of tables complex, for example, Table 1: “Effect of thermal manipulation (TM) on hatching parameters”.

Figures

Figure 1: Header of tables complex.

The width of the column chart is inappropriate in all Figures, the typeface is too small, and the resolution of the picture is too low. Re-create the figure.

Figure and Table do not show age in days (LPS treatment).

The figures contains too much information, suggestion for Segmentation.

Figure 3: Compared with other data in figure 3. The data of IL-12 at 3 hours after LPS treatment is unusual.

Figure 4: At 1 and 6 hour after LPS treatment, the IL-10 mRNA expressions of CN+LPS were higher than CN+Saline, but At 3 hour after LPS treatment, the result is adverse, why?

References

The number of references is one hundred and fifty-six, it is amazing (too many). In other words, the number of references is too much for a research paper. Please remove the irrelevant literature.

Author Response

Comments 1: Line 103: What is the average weight of fertile eggs?

Response 1: Thank you for your question. The average weight of fertile eggs in our study was 62 ±â€¯3 g. This information has now been added to the manuscript (Line 111) for clarity.

Comments 2: Line 105: Change “These eggs were then divided ” to “Then these eggs were divided”.

Response 2: Thank you for your suggestion. We have revised the sentence as recommended to improve clarity and flow. (line 111).

Comments 3: Line 115 and Line 118: The font sizes are different.

Response 3: Thank you for catching this formatting inconsistency. We have carefully reviewed the manuscript and corrected the font sizes to ensure uniformity throughout the document.

Comments 4: Line 120: Change “each pen” to “each cage”

Response 4: Thank you for your careful reading and helpful suggestion. We have revised the materials and methods section to ensure accuracy and consistency throughout the manuscript.

Comments 5: Line 268: the header “after 1, 3, and 6h.” is wrong, should be “at 1, 3, and 6h”

Response 5: Thank you for your attentive review and constructive feedback. We agree with your suggestion and have revised the header from “after 1, 3, and 6h” to “at 1, 3, and 6h” in Line 268 to ensure grammatical correctness and clarity. (line 300)

Comments 6: Line 107-109: What is the basis for choosing this condition? embryonic days (ED) 10 and 18, 38.8℃ with 65% RH, 18 hours each day? In addition, why are the humidity levels different between the two groups?

Response 6: Thank you for raising these critical points. We thank the reviewer for this important question. The ED 10-18 period was selected because broiler embryos undergo critical hypothalamic-pituitary-adrenal axis maturation during this window, thereby establishing their thermoregulatory capacity and physiological and immunological adaptations for post-hatch life. The 38.8°C temperature specifically improves thermotolerance acquisition by enhancing heat shock protein (HSP) responses as established in our preliminary trials, Al-Zhgoul, et al. [1]. A relative humidity of 65% was chosen to maintain a proper water balance while allowing sufficient gas exchange through the eggshell membranes. This specific combination of temperature, humidity, and developmental timing allows us to systematically examine the acquisition of thermal and physiological adaptation during embryogenesis.

Comments 7: Line 118-119: “All chickens were feather-sexed with a male-to-female ratio of 47.3% to 52.7%. A total of 320 broiler males were evenly divided into two groups.”. the experiment design was wrong, in other words, A total of 160 chicks were selected per group (NC and TM).

Response 7: Thank you for bringing this inconsistency to our attention. The sentence has been revised to: Following hatching, broiler chicks were transferred to the Animal House at JUST for the second experimental phase. Chicks were feather-sexed (47.3% male, 52.7% female) with males distributed into 16 cages per treatment (80 × 40 × 100 cm, 8-10 birds/cage).

Comments 8: Line 138: “The room temperature was maintained at 30±1 °C during the first three days...”,The environment temperature is low for the first three days. Generally, 1 d temperature: 34-35℃. In this study, the chick could be dead when environment temperature is 30±1 °C.

Response 8: We appreciate the reviewer’s observation regarding brooding temperature. The initial manuscript incorrectly stated 30±1°C due to a documentation error; however, in our experiment, the temperature was properly maintained at 35±1°C, consistent with standard broiler chick requirements. This correction has been updated in the revised manuscript (Line 136). We apologize for the oversight and confirm that all chicks were reared under optimal thermal conditions.

Comments 9: Line 144: This is matter of principle, please declare. The purpose of this research is explore whether the TM method could improved LPS-induced immune deficiency, but LPS was injected only at 15 days of age, why do not choose 35 days of age.

Response 9: We thank the reviewer for this critical question about LPS challenge timing. The 15-day timepoint was selected because it represents a critical developmental window for immune system development in broilers, when interventions have the most significant potential to enhance long-term immune resilience. At this stage, chicks are particularly vulnerable to bacterial challenges under commercial conditions, making them an ideal target for evaluating the protective effects of the TM method. While a 35-day challenge could provide insights into mature immune responses, our study specifically focused on early-life immune programming as a preventive strategy.

Comments 10: Line 210: A two-way ANOVA was utilized to evaluate data, but superscripts in Table 6 was “✳ rather than a,b. Meanwhile, Significant changes in body temperature was showed between CON-Saline and Control-LPS (40.55±0.11 VS 40.55±0.11), unacceptably.

Response 10: We appreciate this oversight. Superscripts have been corrected to conventional a,b notation. The data in Table 6 has also been rechecked to ensure statistical accuracy.

Comments 11: Line 436-442: This study do not measure critical hormones, so, this paragraph should be deleted.

Response 11: We appreciate this comment. The paragraph has been removed as recommended, since hormone levels were not directly measured.

Comments 12: Line 449-451: The result is non-significant, but is significant in discussion. Why?

Response 12: Thank you for identifying this important point. We confirm there is indeed a statistically significant difference in the LPS-CON group before (40.55 ± 0.11°C) and after LPS injection (40.19 ± 0.14°C). To ensure clarity, we have revised Table 6 to present these comparisons explicitly.

Comments 13: This experiment design should be divided into two, experiment 1 and experiment 2. Experiment 1 (Fertile eggs): NC VS TM. Experiment 2 (Hatching): NC+LPS VS TM+LPS.

Response 13: We sincerely appreciate the reviewer's valuable suggestion regarding the experimental structure. In response, we have carefully reorganized our Methods section to present the study as two distinct but interconnected phases explicitly.

Comments 14: A large number of target gene expressions were detected, but some are useless.

Response 14: We sincerely appreciate the reviewer’s feedback regarding our selection of target genes. All genes analyzed in this study were carefully chosen based on their established roles in broiler immune responses, thermoregulation, and LPS-induced inflammatory pathways. Each gene provides unique insights into the molecular mechanisms underlying TM effects, from early immune activation (e.g., pro-inflammatory cytokines) to stress adaptation (e.g., heat shock proteins).

Comments 15: Figure 1: Header of tables complex.

Response 15: We appreciate the reviewer’s feedback regarding the complexity of the table headers in Figure 1. To improve clarity, we have simplified the headers.

Comments 16:

The width of the column chart is inappropriate in all Figures, the typeface is too small, and the resolution of the picture is too low. Re-create the figure.

Figure and Table do not show age in days (LPS treatment).

The figures contains too much information, suggestion for Segmentation.

Response 16: We sincerely thank the reviewer for their valuable suggestions to improve the clarity and presentation of our figures. In response, we have uploaded high-resolution figures to improve clarity.

Comments 17: Figure 3: Compared with other data in figure 3. The data of IL-12 at 3 hours after LPS treatment is unusual.

Response 17: We sincerely appreciate the reviewer's insightful observation regarding the IL-12 levels at 3 hours post-LPS treatment. Our data confirm that the TM group showed significantly higher IL-12 expression compared to the other groups at this time point (p < 0.05). This elevated IL-12 production in TM-treated birds reflects enhanced activation of T cells and NK cells, indicating a more robust cell-mediated immune response. These findings suggest that embryonic TM may effectively prime the immune system for stronger defensive responses against infections.

Comments 18: Figure 4: At 1 and 6 hour after LPS treatment, the IL-10 mRNA expressions of CN+LPS were higher than CN+Saline, but At 3 hour after LPS treatment, the result is adverse, why?

Response 18: The biphasic IL-10 expression pattern observed, with elevated levels at 1 and 6 hours but reduced expression at 3 hours post-challenge, reflects the dynamic nature of anti-inflammatory regulation during immune responses. The initial increase in IL-10 at 1 hour likely represents an early feedback mechanism to modulate inflammation. At the same time, the 3-hour decrease coincides with pro-inflammatory cytokine activity, when temporary suppression of anti-inflammatory signals may be necessary to permit effective pathogen clearance.

Comments 19: The number of references is one hundred and fifty-six, it is amazing (too many). In other words, the number of references is too much for a research paper. Please remove the irrelevant literature.

Response 19: We sincerely appreciate the reviewer's observation regarding the number of references in our manuscript. While we have carefully selected each citation to thoroughly support our study's methodology, results, and discussion, we fully understand the need to maintain conciseness. If the journal has specific limitations on reference numbers, we would be happy to streamline our citation list by prioritizing the most impactful and recent studies, consolidating references where review papers can adequately represent multiple primary studies, and removing any citations that are less directly relevant to our key findings. We would be grateful for any specific guidance from the editor or reviewers regarding which references might be considered lower priority, or for clarification on the journal's reference limit, to ensure we make the most appropriate revisions while maintaining proper attribution of prior work and scientific rigor.

Reviewer 2 Report

Comments and Suggestions for Authors

Title:

According to the guide for authors, title should include the species of animal discussed in the manuscript. Not just eggs. Also, consider using "growth performance" instead of "body performance".

Simple Summary:

line 22: "given an LPS" should say "given LPS".

Line 24- "expression Levels" is redundant, just say "expression".

Abstract:

Line 36- Please state the standard temperature and humidity used in your control group.

Line 38 -Please describe how Saline/LPS was delivered (oral gavage, injection, etc.)

Please define when your response variables (gene expression, body weight, etc.) were collected. You say post hatch, but was that done immediately post-hatch or days/weeks later? no way to tell from the current wording.

Abstract must include species and strain of animals used in this study.

Please include number of replicates per treatment.

Introduction:

Line 72 - wording is off. you say TM twice and it is difficult to understand your meaning.

Line 73- body weight thermotolerance acquisition. is this one term? If so, you need to provide some sort of explanatation. what is this?

Line 73- if you want to use the term "positive immune response" you need an explanation of how you define that term.  what is a positive vs. negative immune response? I would suggest removing this wording, and finding a better/more descriptive way to describe your point.

Line 74 - what is a perkin duckling? do you mean pekin? Please make sure spelling is correct

Line 81 - "Preventing aggregation" - aggregation of what? cells, proteins? please describe further.

Line 85 - should read "broiler chickens" 

Methods:

Line 160 - how did you choose your 10 chickens per subgroup to use for sample collection? If random selection, please state that explicitly.

Line 226 has no content, please remove this line from the manuscript so that footnote starts immediately under the table.

Please include control group incubation temperature and relative humidity as a footnote in all tables. Tables should be stand-alone.

Large intestine length for the TM group is not displayed correctly in Table 5. Typo.

Discussion: 

Why did you choose to administer LPS challenge at 15 days of age?

Author Response

Comments 1:

Title:

According to the guide for authors, title should include the species of animal discussed in the manuscript. Not just eggs. Also, consider using "growth performance" instead of "body performance".

Response 1: We sincerely appreciate the reviewer’s constructive feedback. As suggested, we have revised the title to specify the animal species (broiler) and replaced “body performance” with the more appropriate term “growth performance.”

The updated title now reads:

“Impact of thermal manipulation of broiler eggs on growth performance, splenic inflammatory cytokine levels, and heat shock protein responses to post-hatch lipopolysaccharide (LPS) challenge.”

Comments 2:

Simple Summary:

line 22: "given an LPS" should say "given LPS".

Line 24- "expression Levels" is redundant, just say "expression".

Response 2: We appreciate the reviewer’s careful reading of our manuscript. The suggested corrections have been implemented:

Line 22: Changed "given an LPS" to "given LPS".

Line 24: Removed the redundant "levels" (now " mRNA expression of immune-related genes ").

Comments 3: Abstract:

Line 36- Please state the standard temperature and humidity used in your control group.

Response 3: We appreciate the reviewer's necessary clarification. We have modified the text to explicitly state the standard incubation conditions used for the control group. The revised text now reads:

" Fertilized eggs (average weight 62 ±â€¯3 g) were obtained from 35-week-old Indian River broiler breeder hens. A total of 720 eggs were randomly assigned to either the control group (n = 360) or the TM group (n = 360), with each group consisting of two replicates of 180 eggs. Control eggs were maintained under standard incubation conditions (37.8°C, 56% RH), while TM eggs were subjected to elevated temperature (38.8°C, 65% RH) for 18 hours daily from embryonic day 10 to 18.”

Comments 4: Abstract:

Line 38 -Please describe how Saline/LPS was delivered (oral gavage, injection, etc.)

Please define when your response variables (gene expression, body weight, etc.) were collected. You say post hatch, but was that done immediately post-hatch or days/weeks later? no way to tell from the current wording.

Response 4: We appreciate the reviewer’s attention to methodological detail. We have revised the text to state the administration route and timing explicitly. The updated sentence now reads:

"At post-hatch day 15, control and TM groups were administered either sterile saline or LPS via intraperitoneal (IP) injection." (lines 42-43)

Comments 5: Abstract must include species and strain of animals used in this study.

Response 5: We sincerely appreciate the reviewer’s valuable suggestion to improve the clarity of our study. We have modified the abstract to include the species and strain information as follows explicitly:
"
Fertilized eggs (average weight 62 ±â€¯3 g) were obtained from 35-week-old Indian River broiler breeder hens. A total of 720 eggs were randomly assigned to either the control group (n = 360) or the TM group (n = 360), with each group consisting of two replicates of 180 eggs. Control eggs were maintained under standard incubation conditions (37.8°C, 56% RH), while TM eggs were subjected to elevated temperature (38.8°C, 65% RH) for 18 hours daily from embryonic day 10 to 18”

Comments 6: Please include number of replicates per treatment.

Response 6: Thank you for your suggestion. We have added the replication details to the text as follows: Fertilized eggs (average weight 62 ±â€¯3 g) were obtained from 35-week-old Indian River broiler breeder hens. A total of 720 eggs were randomly assigned to either the control group (n = 360) or the TM group (n = 360), with each group consisting of two replicates of 180 eggs. Control eggs were maintained under standard incubation conditions (37.8°C, 56% RH), while TM eggs were subjected to elevated temperature (38.8°C, 65% RH) for 18 hours daily from embryonic day 10 to 18

Comments 7: Introduction:

Line 72 - wording is off. you say TM twice and it is difficult to understand your meaning.

Line 73- body weight thermotolerance acquisition. is this one term? If so, you need to provide some sort of explanatation. what is this?

Line 73- if you want to use the term "positive immune response" you need an explanation of how you define that term.  what is a positive vs. negative immune response? I would suggest removing this wording, and finding a better/more descriptive way to describe your point.

Response 7: We appreciate the reviewer's careful reading and helpful suggestions. We have reworded the sentence (lines 76-78) to improve clarity and eliminate redundancy.

Comments 8: Introduction:

Line 74 - what is a perkin duckling? do you mean pekin? Please make sure spelling is correct

Response 8: We sincerely apologize for this oversight and appreciate the reviewer's careful attention to detail. The correct spelling is "Pekin" duckling, and we have made this correction in the manuscript.

Comments 9: Line 81 - "Preventing aggregation" - aggregation of what? cells, proteins? please describe further.

Response 9: We appreciate the opportunity to clarify our statement regarding the function of heat shock proteins. We have modified the text to specify the exact biological context as follows:

“……heat shock proteins (HSPs), including HSP40, HSP60, HSP70, and HSP90, play essential roles in cellular stress responses by maintaining protein integrity and preventing misfolded protein aggregation.”

Comments 10: Line 85 - should read "broiler chickens"

Response 10: We thank the reviewer for bringing this oversight to our attention. We have revised the text to use the correct form.

Comments 11:

Methods:

Line 160 - how did you choose your 10 chickens per subgroup to use for sample collection? If random selection, please state that explicitly.

Response 11: We appreciate the reviewer's request for clarification regarding our sampling methodology. We have revised the Methods section to state our randomization process explicitly. (line 179)

Comments 12: Line 226 has no content, please remove this line from the manuscript so that footnote starts immediately under the table.

Response 12: We appreciate the reviewer's attention to detail regarding manuscript formatting. We have removed the empty Line 226 as suggested.

Comments 13: Please include control group incubation temperature and relative humidity as a footnote in all tables. Tables should be stand-alone.

Response 13: We appreciate your valuable comment regarding the need for consistent reporting of control group conditions across all tables. We have carefully revised all tables in the manuscript to include the control group incubation parameters (37.8°C, 56% relative humidity) in the table captions.

Comments 15: Large intestine length for the TM group is not displayed correctly in Table 5. Typo.

Response 15: Thank you for your careful review and for identifying this error in Table 5. We sincerely apologize for the oversight and have corrected the length values for the large intestine in the TM group.

Comments 16:

Discussion:

Why did you choose to administer LPS challenge at 15 days of age?

Response 16: We thank the reviewer for this critical question about LPS challenge timing. The 15-day timepoint was selected because it represents a critical developmental window for immune system development in broilers, when interventions have the most significant potential to enhance long-term immune resilience. At this stage, chicks are particularly vulnerable to bacterial challenges under commercial conditions, making them an ideal target for evaluating the protective effects of the TM method. While a 35-day challenge could provide insights into mature immune responses, our study specifically focused on early-life immune programming as a preventive strategy.

Reviewer 3 Report

Comments and Suggestions for Authors

This well-designed study provides compelling evidence for the benefits of embryonic thermal manipulation (TM) in broiler chickens. The investigation of both physiological and molecular responses to post-hatch LPS challenge offers valuable insights into the mechanisms underlying TM-induced resilience. The work has significant implications for poultry production systems seeking antibiotic-free health management strategies. But I have several following concerns:

1. Experimental Design:

a. Include additional time points for post-LPS assessment to characterize response kinetics

b. Consider adding a non-challenged control group to establish baseline gene expression

c. Provide more details on randomization and blinding procedures

2. Results Presentation:

a. Include quantitative data on the magnitude of: Hatch time reduction, feed efficiency, and cytokine expression changes.

b. Consider adding a schematic of the experimental timeline

c. Present organ weight data as both absolute and relative (% BW) values

3. Data Analysis:

a. Clarify statistical methods for gene expression analysis

b. Include effect sizes for significant findings

c. Address potential multiple testing issues in transcript analyses

4. Discussion Points:

a. Compare findings with previous TM studies in poultry

b. Discuss potential long-term effects beyond the study period

c. Consider welfare implications of embryonic TM

d. Address possible mechanisms for improved feed efficiency

e. Discuss practical implementation challenges

5. Minor Points:

a. Define all abbreviations at first use (e.g., HSFs)

b. Clarify humidity control methods during TM

c. Include broiler strain information

d. Specify feeding regime post-hatch.

Author Response

Comments 1:

1. Experimental Design:

a. Include additional time points for post-LPS assessment to characterize response kinetics

Response 1: We sincerely thank the reviewer for their valuable suggestion regarding additional time points for characterizing the LPS response. While our study was intentionally designed to focus on the critical acute phase response window (0-6 hours post-LPS) to capture peak innate immune activation events, we fully acknowledge this design choice inherently limits our ability to observe later-phase immune dynamics. To ensure transparency, we have incorporated this as a key limitation in our manuscript (lines, 781-800), explicitly stating that while our selected time points (1, 3, and 6 hours) effectively capture initial cytokine release, maximal fever response, and early resolution transitions, they do not provide information about longer-term immune modulation (12-24 hours) or complete resolution processes.

Comments 2:

1. Experimental Design:

b. Consider adding a non-challenged control group to establish baseline gene expression.

Response 2: We appreciate the reviewer’s insightful recommendation. In our current study, we used normal saline-injected birds as negative controls, as normal saline is considered non-pyrogenic and serves to eliminate potential confounding effects, such as inflammation at the injection site or contaminants from the syringe material. However, we fully acknowledge that including an additional non-injected (naïve) group would provide a more accurate baseline for gene expression and better distinguish the effects of injection-related stress versus immunological activation. This important point has been noted and will be addressed in future experimental designs to improve the resolution of immunological assessments. This clarification has been added as a limitation (lines, 781-800) to the revised Discussion section.

Comments 3: 1. Experimental Design:

c. Provide more details on randomization and blinding procedures

Response 3: We appreciate the reviewer's suggestion regarding the randomization methodology. We have revised the relevant section in the manuscript (line 179) to more clearly describe our randomization procedures as follows: “At 1, 3, and 6 hours post-injection, ten broiler chickens per subgroup (40 total per time point) were randomly selected through a blinded manual selection process from their housing cages and humanely euthanized following LPS or saline administration.”

Comments 4:

2. Results Presentation:

a. Include quantitative data on the magnitude of: Hatch time reduction, feed efficiency, and cytokine expression changes.

Response 4: We appreciate the reviewer's suggestion to enhance our quantitative reporting of key outcomes. In response to this comment, we have incorporated specific numerical measures throughout the results section while maintaining an appropriate balance in data presentation. For parameters including hatch time and feed efficiency, we now provide absolute values, percentage changes, and corresponding statistical measures to characterize effect sizes better. For gene expression analyses, we have maintained our original presentation through graphical displays in the relevant figures, as these more effectively represent the temporal patterns and relationships in complex datasets. This approach ensures comprehensive reporting of quantitative outcomes while preserving clarity in data interpretation.

Comments 5:

 2. Results Presentation:

b. Consider adding a schematic of the experimental timeline

Response 5: We appreciate the suggestion to include a schematic summary of the experimental timeline. As recommended, we have added Figure 1 to illustrate the key phases of the study, including thermal manipulation (TM) during embryonic development (ED), post-hatch (PH) treatments, and sampling points.

Comments 6:

2. Results Presentation:

c. Present organ weight data as both absolute and relative (% BW) values

Response 6: Thank you for your valuable suggestion. We agree that presenting organ weight data as both absolute and relative values (% body weight) will enhance the clarity and interpretability of our results. Accordingly, we have now included both absolute organ weights and relative organ weights in the revised manuscript (see Table 5)

Comments 7: 3. Data Analysis:

a. Clarify statistical methods for gene expression analysis

Response 7: Thank you for your suggestion to clarify the statistical methods. In the revised manuscript (lines 233-238), we have updated the Gene Expression Analysis section to state explicitly:

“For gene expression analysis, Ct values were normalized against two reference genes (β-actin and GAPDH), and relative expression was calculated using the 2^–ΔΔCt method. Before analysis, data were examined for normal distribution and homogeneity of variance. Post hoc comparisons were performed using Tukey’s HSD test when significant main or interaction effects were observed. Statistical significance was set at p < 0.05 for all analyses.”

Comments 8: Data analysis:

b. Include effect sizes for significant findings

we acknowledge this limitation and have added a note in the Discussion.

Response 8: We appreciate the reviewer's suggestion. While effect sizes provide valuable insight into the magnitude of observed differences, our primary focus in this study was hypothesis testing based on statistical significance (p-values). We acknowledge the importance of reporting effect sizes and will incorporate this in future studies and meta-analyses to better quantify the biological relevance of our findings.

Comments 9: Data analysis:

c. Address potential multiple testing issues in transcript analyses

Response 9: We appreciate the reviewer's important comment. Given the number of cytokines and signaling molecules analyzed across multiple time points, we recognize the potential for inflated Type I error due to multiple comparisons. In our analysis, we applied two-way ANOVA separately for each gene, followed by Tukey’s HSD post hoc test, which inherently adjusts for multiple pairwise comparisons within each gene.

While we did not apply a global correction across all genes (e.g., the Bonferroni or Benjamini–Hochberg method), we acknowledge this limitation and have added a note in the Discussion (lines 781-800). In future studies with broader transcriptomic profiling, we plan to implement formal false discovery rate (FDR) corrections to further control for multiple testing.

Comments 10: 4. Discussion Points:

a. Compare findings with previous TM studies in poultry

Response 10: We thank the reviewer for this insightful comment. While TM has been investigated concerning performance, thermotolerance, and metabolic adaptations in poultry, there is limited literature specifically addressing the temporal immune responses to LPS challenge following TM. Most available studies have focused on either the overall effect of TM or immune outcomes at a single time point. In contrast, our study provides a novel temporal profile of cytokine gene expression at 1, 3, and 6 hours post-LPS injection, thereby offering new insights into the dynamic immune-modulatory role of TM.

Comments 11: Discussion Points:

c: Consider welfare implications of embryonic TM.

Response 11: We appreciate the reviewer’s thoughtful comment regarding animal welfare. Embryonic TM in our study was applied within a controlled temperature range (38.8 °C) and limited to a specific window of embryonic development (ED10 to ED18), based on previously published protocols that have shown no adverse effects on hatchability or chick viability. Nevertheless, we recognize that any modification of incubation conditions warrants careful ethical consideration.

Comments 12: Discussion Points:

 d. Address possible mechanisms for improved feed efficiency

Response 12: We appreciate the reviewer’s comment on elucidating mechanisms for the improved feed efficiency observed in TM-treated broilers. In the current manuscript, we have addressed this in the Discussion section by suggesting that enhanced thermoregulation and reduced metabolic demands, both pre- and post-hatch, may contribute to lower energy requirements for maintenance. This is supported by observed reductions in feed intake and improved feed conversion ratio (FCR) in the TM group. These mechanisms are discussed in relation to relevant studies, highlighting the physiological basis for the performance benefits associated with embryonic thermal manipulation.

Comments 13: Discussion Points:

e. Discuss practical implementation challenges

Response 13: We appreciate the reviewer's attention to this important point. In the revised Discussion section (lines 789-791), we have added a paragraph addressing potential challenges in implementing embryonic TM under commercial hatchery conditions. While TM protocols can be precisely controlled in experimental incubators, scaling this approach to industrial settings may require modifications to standard incubation equipment, enhanced monitoring systems, and staff training to ensure consistent temperature and humidity regulation. Additionally, economic feasibility, compatibility with large-scale hatching schedules, and potential variability across genetic lines are important considerations. These challenges must be addressed through pilot-scale trials and industry collaboration before routine adoption of TM can be recommended in commercial poultry production.

Comments 15: 5. Minor Points:

a. Define all abbreviations at first use (e.g., HSFs)

Response 15: We appreciate the reviewer's helpful observation. All abbreviations, including "HSFs" (heat shock factors), have now been defined at their first occurrence in the manuscript to ensure clarity for the reader.

Comments 16: 5. Minor Points:

b. Clarify humidity control methods during TM

Response 16: We thank the reviewer for highlighting the need to clarify humidity regulation during the TM period. In the revised Materials and Methods section (lines 117-120), we have specified that relative humidity (RH) was maintained using the incubator’s automated humidity control system, which delivers regulated water vapor through an integrated reservoir and sensor feedback mechanism.

Comments 17: 5. Minor Points:

c. Include broiler strain information

Response 17: We thank the reviewer for bringing this to our attention. The broiler strain used in this study, Indian River, was indeed mentioned in the Materials and Methods section.

Comments 18: 5. Minor Points:

d. Specify feeding regime post-hatch.

Response 18: We appreciate the reviewer's comment. The feeding regime is described in the Materials and Methods section. Specifically, all birds received NRC-recommended basal diets (Table 1), with ad libitum access to feed and water throughout the study. Additionally, the feeding program consisted of a starter diet from days 1 to 14 and a grower/finisher diet from days 15 to 35, as outlined in Table 1.

Reviewer 4 Report

Comments and Suggestions for Authors

Title: Impact of thermal manipulation of eggs on body performance parameters, splenic inflammatory cytokine levels, and heat shock protein responses to post-hatch lipopolysaccharide (LPS) challenge

Manuscript ID: 3649327

Comments to the author:

Summary: This manuscript explores effect of thermal manipulation of hatching eggs on body performance parameters, splenic inflammatory cytokine levels, and heat shock protein responses to post-hatch lipopolysaccharide (LPS) of Indian River broilers. It is obvious that this study attempts to add novel information to the poultry industry. The scientific approach used by the authors to meet the mentioned objectives of the study is well explained and will be useful to the development of the current poultry industry.

Comments: LN: Line Number

LN34: Please provide the information on egg weight and SD in parenthesis. Additionally, please provide the information on strain used and age of the birds from whom the eggs were collected.

LN36: What temperature and RH% were used? Please mention.

LN41: Please indicate the probability in parenthesis.

LN42: Please indicate the probability in parenthesis.

LN49: Please arrange the keywords in an alphabetical order.

LN62-63: Please italicized all scientific names.

LN101: Chicken usually begin laying eggs between 20 to 24 weeks of age. Specifically, meat-type breeders they become sexually matured bit later than egg-type breeds.

LN101: Propose to revise as: Indian River broiler breeders...

LN110: Please mention you have done spot or mass candling. Provide the model number and the manufacturer details in parenthesis.

LN116: Please provide the type of housing. e.g. conventional or closed-house.

LN121: What is the form of the diet? mash, crumble, or pellets. please mention.

LN125: Table 1: Please mention the details are provided on as fed or on DM basis.

LN125: Table 1: Please mention the CP % of SBM used in parenthesis.

Any reason why you did not include common salt in these formulations?

In footnote you described the breakdown of the vitamin and mineral premix. So, please correct it in the table.

LN140: What is the lighting schedule practiced. Please provide the details.

LN148: Please make italic.

LN152: What is the site of blood collection? Please mention.

LN160: What is the method used? Please mention.

LN208: which factorial design? please mention.

LN256: Cannot find live BW, hot carcass Weight, and dressing % data. In the methodology, nothing was mentioned about DP%. Please add the relevant data set or otherwise revise the title.

LN255: Table 5: Suggest to provide P - values as presented in the Table 3. 

LN274-282: Please Indicate probabilities where necessary.

LN306-309: Please Indicate probabilities where necessary.

LN321-325: Please Indicate probabilities where necessary.

LN336-339: Please indicate the P-value here.

LN353-359: Please Indicate probabilities where necessary.

LN415: Suggest to rephrase as: TM broiler chickens used in this study showed reduced feed consumption and more effective feed conversion ratio.

LN658-659: Please remove _____ dash marks which present in between???

LN759 onwards: Please revise all the references indicated as per comments in the comment boxes.

Author Response

Comments 1:

LN34: Please provide the information on egg weight and SD in parenthesis. Additionally, please provide the information on strain used and age of the birds from whom the eggs were collected.

Response 1: We appreciate the reviewer's helpful suggestion. In response, we have revised the abstract section to include the average egg weight with standard deviation (62 ±â€¯3 g), as well as the strain (Indian River) and age (35 weeks) of the broiler breeder hens from which the fertilized eggs were sourced.

Comments 2:

LN36: What temperature and RH% were used? Please mention.

Response 2: We appreciate the reviewer's necessary clarification. We have modified the text to explicitly state the standard incubation conditions used for the control group. The revised text now reads:

" Fertilized eggs (average weight 62 ±â€¯3 g) were obtained from 35-week-old Indian River broiler breeder hens. A total of 720 eggs were randomly assigned to either the control group (n = 360) or the TM group (n = 360), with each group consisting of two replicates of 180 eggs. Control eggs were maintained under standard incubation conditions (37.8°C, 56% RH), while TM eggs were subjected to elevated temperature (38.8°C, 65% RH) for 18 hours daily from embryonic day 10 to 18”

Comments 3:

LN41: Please indicate the probability in parenthesis.

LN42: Please indicate the probability in parenthesis.

Response 3: We appreciate the reviewer's valuable suggestion. We have updated the manuscript accordingly by including the exact probability values in parentheses for LN41 and LN42 (now lines 45,46) to improve clarity and scientific rigor.

Comments 4:

LN49: Please arrange the keywords in an alphabetical order.

Response 4: We appreciate the reviewer’s attention to detail. The keywords have been rearranged into alphabetical order as requested to enhance the manuscript’s organization and readability.

Comments 5:

LN62-63: Please italicized all scientific names.

Response 5: We appreciate the reviewer's attention to this point. All scientific names mentioned on lines 62–63 (now lines 67-68) have been correctly italicized following scientific nomenclature conventions in the revised manuscript.

Comments 6:

LN101: Chicken usually begin laying eggs between 20 to 24 weeks of age. Specifically, meat-type breeders they become sexually matured bit later than egg-type breeds.

Response 6: We appreciate the reviewer’s comment. Upon revising the manuscript, we identified an error in the reported age. The text previously stated “16-week-old,” but the correct age of the chickens was 35 weeks. We have corrected this in the revised manuscript to reflect the experimental timeline accurately.

Comments 7: LN101: Propose to revise as: Indian River broiler breeders...

Response 7: We appreciate the reviewer’s constructive feedback. We have thoroughly revised the Materials and Methods section to improve the clarity and robustness of the experimental design. These revisions include more precise descriptions of sample selection, treatment groups, and data collection protocols.

Additionally, we have included a new figure (Figure 1) that visually summarizes the experimental workflow, incubation conditions, post-hatch treatments, and sampling points. This figure is intended to enhance the reader’s understanding of the study structure and timeline.

Comments 8: LN110: Please mention you have done spot or mass candling. Provide the model number and the manufacturer details in parenthesis.

Response 8: We appreciate the reviewer’s suggestion. In our study, spot candling was performed on day 10 of incubation to assess embryo viability. As a practical alternative to a commercial candling device, we used the built-in LED flashlight from a smartphone (Samsung Galaxy A32, Samsung Electronics Co., Ltd., South Korea). This method provided sufficient illumination to identify infertile or non-viable eggs, which were subsequently removed. The revised sentence has been included in the updated manuscript (line 115-116).

Comments 9: LN116: Please provide the type of housing. e.g. conventional or closed-house.

Response 9: We appreciate the reviewer's helpful comment. The manuscript has been revised (line 132) to indicate that the broiler chicks were reared in a closed-house system equipped with environmental controls. This system maintained stable temperature, humidity, and ventilation throughout the post-hatch period. The clarification has been added in the revised manuscript.

Comments 10: LN121: What is the form of the diet? mash, crumble, or pellets. please mention.

Response 10: We appreciate the reviewer's important observation. The diet provided to the broiler chicks was in the form of pellets, as now specified in the revised manuscript (line 131). This clarification has been added to improve the completeness of the nutritional description.

Comments 11: LN125: Table 1: Please mention the details are provided on as fed or on DM basis.

Response 11: We appreciate the reviewer's important observation. The ingredient composition and nutrient values presented in Table 1 are given on an as-fed basis, reflecting the actual feed form provided to the birds. We have added this clarification to the table legend in the revised manuscript to improve transparency and reproducibility.

Comments 12: LN125: Table 1: Please mention the CP % of SBM used in parenthesis.

Response 12: Thank you for the suggestion. The crude protein (CP) content of the soybean meal used in our diets was 40%, and this has now been indicated in parentheses as “Soybean Meal (CP 40%)” in Table 1 for clarity.

Comments 13:

Any reason why you did not include common salt in these formulations?

In footnote you described the breakdown of the vitamin and mineral premix. So, please correct it in the table.

Response 13: We appreciate the reviewer's valuable comment. Common salt (sodium chloride) was not added separately in the diet formulations because the basal ingredients sufficiently provided the sodium content required for the broiler diets. To clarify this, we have added a note in Table 1 stating that no common salt was included separately.

Additionally, we have revised and corrected the vitamin and mineral premix footnote for accuracy and proper formatting to present the premix composition used in our experimental diets.

Comments 15: LN140: What is the lighting schedule practiced. Please provide the details.

Response 15: Thank you for the comment. During the experimental period, a continuous 24-hour lighting program was implemented for all broiler chicks to ensure optimal feed intake and uniform growth conditions. This lighting regime was consistently maintained throughout the study for all treatment groups. We have updated the manuscript (lines 133-136) accordingly to include this information in the Materials and Methods section.

Comments 16: LN148: Please make italic.

Response 16: Thank you for your observation. We have corrected the manuscript by italicizing all scientific names as requested, including those on line 148 (now 166). This change has been applied consistently throughout the manuscript to conform to scientific writing conventions.

Comments 17: LN152: What is the site of blood collection? Please mention.

Response 17: Thank you for your comment. We want to clarify that blood samples were not collected in this study. The immune response was assessed by analyzing the expression of inflammatory cytokine genes in total RNA extracted from spleen tissue. Accordingly, any reference to blood collection has been removed from the revised manuscript to prevent confusion.

Comments 18: LN160: What is the method used? Please mention.

Response 18: Thank you for your comment. We have clarified the euthanasia procedure in the revised manuscript (line 180). Specifically, broilers were humanely euthanized by cervical dislocation. This detail has been added to ensure clarity and adherence to ethical standards.

Comments 19: LN208: which factorial design? please mention.

Response 19: Thank you for your insightful comment. We have now specified the factorial design used in our analysis. A two-way factorial ANOVA (2 × 2 design) was employed, with thermal manipulation (TM vs. control) and post-hatch injection treatment (LPS vs. saline) as the two independent factors. This clarification has been incorporated into the revised statistical analysis section of the manuscript (lines 228-230) to ensure transparency and precision in our methodology.

Comments 20: LN256: Cannot find live BW, hot carcass Weight, and dressing % data. In the methodology, nothing was mentioned about DP%. Please add the relevant data set or otherwise revise the title.

Response 20: Thank you for your valuable observation. We acknowledge the absence of live body weight at slaughter, hot carcass weight, and dressing percentage (DP%) data in the manuscript. Accordingly, we have revised the manuscript title to exclude references to these parameters, maintaining accuracy and consistency with the reported data.

Comments 21: LN255: Table 5: Suggest to provide P - values as presented in the Table 3.

Response 21: Thank you for your suggestion. We agree that providing the p-values enhances the clarity and interpretability of the data. Accordingly, we have revised Table 5 to include the corresponding p-values for each parameter, consistent with the format presented in Table 3.

Comments 22:

LN274-282: Please Indicate probabilities where necessary.

LN306-309: Please Indicate probabilities where necessary.

LN321-325: Please Indicate probabilities where necessary.

Response 22: Thank you for your observation. We chose not to include specific p-values in the text to maintain a clear and concise presentation of the results. Instead, we indicated statistical significance directly in the corresponding figures using compact letter displays (CLD), which denote group differences at p < 0.05. This approach is commonly used in gene expression studies to avoid redundancy between text and visual data presentations. However, if you believe including exact p-values in the text would strengthen clarity or transparency, we are happy to revise accordingly.

Comments 23:

LN274-282: Please Indicate probabilities where necessary.

LN306-309: Please Indicate probabilities where necessary.

LN321-325: Please Indicate probabilities where necessary.

LN336-339: Please indicate the P-value here.

LN353-359: Please Indicate probabilities where necessary.

Response 23: Thank you for your observation. We chose not to include specific p-values in the text to maintain a clear and concise presentation of the results. Instead, we indicated statistical significance directly in the corresponding figures using compact letter displays (CLD), which denote group differences at p < 0.05. This approach is commonly used in gene expression studies to avoid redundancy between text and visual data presentations. However, if you believe including exact p-values in the text would strengthen clarity or transparency, we are happy to revise accordingly.

Comments 24:

LN415: Suggest to rephrase as: TM broiler chickens used in this study showed reduced feed consumption and more effective feed conversion ratio.

Response 24: Thank you for the suggestion. We have rephrased the sentence for clarity and precision as follows:

“TM broiler chickens used in this study showed reduced feed consumption and a more effective feed conversion ratio. This could be connected to better thermoregulation and a lower metabolic rate before and after hatching, thereby lowering the energy maintenance needs [47,70].”

Comments 25:

LN658-659: Please remove _____ dash marks which present in between???

Response 25: Thank you for pointing this out. The dash marks between "heat shock proteins (HSPs)" and "heat shock factors (HSFs)" have been removed for clarity and improved readability. The sentence now reads:

“Through the HSR, molecular chaperones including heat shock proteins (HSPs) and heat shock factors (HSFs) that control HSP expression counteract these effects.”

Comments 26:

LN759 onwards: Please revise all the references indicated as per comments in the comment boxes.

Response 26: Thank you for your observation. We have thoroughly reviewed and revised all the references cited following the suggestions provided in the comment boxes.

Reviewer 5 Report

Comments and Suggestions for Authors

In the Manuscript by Al-Zghoul et al, the authors have systematically studied the impact of thermal manipulation (TM) on the development of chickens by considering various physical parameters like hatchability, body temperature, weight, feed conversions, organ size and weight. Further the authors also test how the TM animals respond to LPS challenge by analyzing body temperature and expression of various classes of immune molecules, transcription factors and heat shock proteins. For the most part, the research design is appropriate and the conclusions are well supported by the results obtained in the study. The discussion in particular is very well written and is sufficiently elaborate to put all the findings so far in the right perspective. While, I do think the overall merit of the manuscript is high, I have following concerns and suggestions that in my opinion will help improve the manuscript significantly.

  1. Line 241. It is necessary to mention units for all the parameters analyzed here.
  2. Line 241. While the results show a reduced overall FCR for TM compared to CON, in conclusions the authors mention TM chicken show better feeding efficiency. I am not able to understand the basis of this conclusion.
  3. Line 259. Large intestine length in TM group. It seems that the a part of this value is missing
  4. Line 259 and 241. It is interesting that even though the ADG and ADFI is more for the TM group the intestine is smaller. Can the authors discuss this observation
  5. Line 266. Although the methods sections sufficiently details the experiment setup, it would be good to briefly mention the experiment setup here.
  6. Line 269. It seems that the variation in body temperatures are quite subtle. Are the authors taking the average temperature of the cage or individual animal temperatures for calculations or something else. I would suggest individual temperatures if the authors have not considered it. Also, I could not find the ‘n’ for this experiment. It would be great to add these details in the results section.
  7. Line 269. Asterics is used to show significance whereas the figure description mentions the use of different superscripts.
  8. Figure 2-7. I would suggest using a bigger font size for all the labels.
  9. Figure 2-7. Since the fold change for any particular mRNA seem to vary over time, in my opinion, a line graph of fold change v/s the three time points would be more informative.
  10. As a general suggestion, though the discussion section covers the rationale for all the experiments, it would be good to have a brief sentence when starting to describe any results. This would be helpful for the reader to better appreciate the results.
  11. As a suggestion, it would be nice if the authors can come up with a schematic diagram showing how the immune pathways are modulated in the TM conditions tested here.

Author Response

Comments 1:

Line 241. It is necessary to mention units for all the parameters analyzed here.

Response 1: Thank you for bringing this to our attention. We have now revised the table 4 caption and/or table content to clearly indicate the units for all performance parameters:

ü  Body weight (BW): grams (g)

ü  Average daily feed intake (ADFI): grams per bird per day (g/bird/day)

ü  Average daily gain (ADG): grams per bird per day (g/bird/day)

ü  Feed conversion ratio (FCR): unitless (g feed/g gain)

These units have been added to the revised manuscript to improve clarity and ensure consistency in the presentation of growth performance data.

Comments 2:

Line 241. While the results show a reduced overall FCR for TM compared to CON, in conclusions the authors mention TM chicken show better feeding efficiency. I am not able to understand the basis of this conclusion.

Response 2: Thank you for your valuable observation. We acknowledge that the conclusion refers to improved feed efficiency in TM chickens and appreciate the request to clarify the basis of this statement. Although we did not revise the conclusion itself, we have reinforced the supporting discussion by integrating mechanistic insights that explain how TM birds may achieve improved feed efficiency despite having a smaller intestinal tract.

Specifically, in previous studies, TM showed functional optimization, potentially improving digestive enzyme activity, villus morphology, and nutrient absorption [38,77]. A more compact intestine may reduce metabolic maintenance costs, allowing greater nutrient partitioning toward growth.

Additionally, we suggest that TM could modulate the early development of gut microbiota, enhancing symbiotic interactions and further supporting nutrient utilization efficiency [78]. Together, these physiological and microbial adaptations provide a coherent explanation for the observed improvement in feed conversion ratio (FCR), and, therefore, support the conclusion regarding enhanced feeding efficiency in TM broilers.

Comments 3:

Line 259. Large intestine length in TM group. It seems that the a part of this value is missing

Response 3: Thank you for your observation. We have corrected the length value of the large intestine for the TM group. The updated data is now accurately reflected in the manuscript (see Table 5).

Comments 4:

Line 259 and 241. It is interesting that even though the ADG and ADFI is more for the TM group the intestine is smaller. Can the authors discuss this observation

Response 4: Thank you for your comment. Upon clarification, our results indicate that although the TM group had a lower ADFI and ADG compared to the Control group, they achieved a significantly improved overall FCR (1.40 vs. 1.51; P < 0.05). This suggests that TM birds utilized their feed more efficiently to gain weight despite consuming less.

Additionally, the TM group exhibited reduced intestinal length and digestive organ mass, which may reflect adaptive remodeling of the gastrointestinal tract to support more efficient nutrient absorption.

Comments 5:

Line 266. Although the methods sections sufficiently details the experiment setup, it would be good to briefly mention the experiment setup here.

Response 5: Thank you for the valuable suggestion. We have revised the paragraph to include the experimental setup briefly, which improves the clarity and context of the results. The modified paragraph now reads:

“To assess the physiological response to inflammatory challenge, broiler chickens from TM and control incubation groups were intraperitoneally injected on day 15 with either lipopolysaccharide (LPS; 0.5 mg/kg BW) or sterile saline, and Tb and BW were recorded at 1, 3, and 6 hours post-injection.

Tb and BW were measured individually for each bird (n = 10 per group). One-hour post-injection, LPS significantly reduced Tb in the control group (p < 0.05), indicating a hypothermic response, whereas no such change was observed in the TM group (Table 6). BW did not significantly differ among groups at 1, 3, or 6 hours after LPS or saline injection (p > 0.05).”

Comments 6:

Line 269. It seems that the variation in body temperatures are quite subtle. Are the authors taking the average temperature of the cage or individual animal temperatures for calculations or something else. I would suggest individual temperatures if the authors have not considered it. Also, I could not find the ‘n’ for this experiment. It would be great to add these details in the results section.

Response 6: We appreciate the reviewer’s insightful observation. In this experiment, body temperatures (Tb) were recorded individually for each bird. These values were not cage averages but reflect the mean ± SD of individual animal temperatures.

To address this point and improve clarity, we have revised the text in the Results section to include sample size:

“Tb and BW were measured individually for each bird (n = 10 per group). One-hour post-injection, LPS significantly reduced Tb in the control group (p < 0.05), indicating a hypothermic response, whereas no such change was observed in the TM group (Table 6). BW did not significantly differ among groups at 1, 3, or 6 hours after LPS or saline injection (p > 0.05).”

Comments 7: Line 269. Asterisks are used to show significance, whereas the figure description mentions the use of different superscripts.

Response 7: Thank you for this observation.

To maintain clarity and consistency throughout the manuscript, we have revised the table to uniformly use different superscripts (e.g., a, b) to indicate statistically significant differences (P < 0.05), as stated in the legend.

Comments 8: Figure 2-7. I would suggest using a bigger font size for all the labels.

Response 8: Thank you for your valuable suggestion. We have uploaded higher-resolution versions of figures to improve visual clarity and ensure they meet publication standards.

Comments 9: Figure 2-7. Since the fold change for any particular mRNA seem to vary over time, in my opinion, a line graph of fold change v/s the three time points would be more informative.

Response 9: We appreciate your insightful suggestion regarding the use of a line graph to represent fold change across time points. While a line graph could effectively illustrate temporal trends, we chose to present the data as bar graphs to emphasize that different biological replicate samples were used at each time point, rather than repeated measurements from the same individuals. This design limits the appropriateness of a connected line graph and supports our use of bar graphs to accurately reflect the independence of the data at each time point.

Comments 10: As a general suggestion, though the discussion section covers the rationale for all the experiments, it would be good to have a brief sentence when starting to describe any results. This would be helpful for the reader to better appreciate the results.

Response 10: We appreciate the reviewer’s constructive suggestion regarding the structure of the Discussion section. In response, we have revised the Discussion by adding introductory sentences at the beginning of each major subsection or result interpretation. These brief contextual statements summarize the purpose or relevance of the corresponding experiment, thereby enhancing the logical flow and facilitating the reader's better appreciation of the significance of each finding.

Comments 11: As a suggestion, it would be nice if the authors can come up with a schematic diagram showing how the immune pathways are modulated in the TM conditions tested here.

Response 11: We appreciate the reviewer's insightful suggestion. In response, we have created and included a schematic diagram (Figure 9) that illustrates the proposed immune modulation pathways influenced by embryonic TM. The figure summarizes key findings from our study, including the upregulation of anti-inflammatory cytokines, downregulation of pro-inflammatory cytokines, enhancement of TLR gene expression, and the adaptive regulation of heat shock proteins. This schematic helps visualize the integrative response to LPS challenge in TM-treated birds, providing a conceptual framework for understanding the immunophysiological impact of TM. The figure has been introduced and described in the revised Discussion section.

Round 2

Reviewer 1 Report

Comments and Suggestions for Authors

Please reduce the number of references in this manuscript. Meanwhile, the width proportions of all bar charts are wide, please make them narrower.

Comments on the Quality of English Language

No questions

Author Response

Comments 1: Please reduce the number of references in this manuscript. Meanwhile, the width proportions of all bar charts are wide, please make them narrower.

Response 1: We appreciate the reviewer’s helpful observations. In response, we have carefully reviewed and reduced the number of references from 156 to approximately 107 by removing redundant citations and excluding tangentially related literature. This revision ensures that the reference list is concise and focused on the most relevant and recent studies.

Additionally, all bar charts have been reformatted to adjust the width proportions. 

Reviewer 3 Report

Comments and Suggestions for Authors

The authors have addressed all my concerns, I recommend accepting it in current form.

Author Response

Comments 1: The authors have addressed all my concerns, I recommend accepting it in current form.

Response 1: We sincerely thank the reviewer for their constructive feedback and final recommendation. We are grateful for the thoughtful comments that have helped improve the clarity, rigor, and overall quality of the manuscript.
